# Tonic extracellular glutamate and ischaemia: glutamate antiporter system xc⁻ regulates anoxic depolarization in hippocampus

Bradley S. Heit[1,2], Alex Chu[2], Abhay Sane[2], David E. Featherstone[3], Thomas J. Park[3] 🆔 and John Larson[2,3] 🆔

[1] *Graduate Program in Neuroscience, University of Illinois at Chicago, Chicago, IL, USA*
[2] *Department of Psychiatry, University of Illinois at Chicago, Chicago, IL, USA*
[3] *Department of Biological Sciences, University of Illinois at Chicago, Chicago, IL, USA*

Handling Editors: Katalin Toth & Samuel Young

The peer review history is available in the Supporting Information section of this article (https://doi.org/10.1113/JP283880#support-information-section).

**Bradley Stavros Heit** is an electrophysiologist at Northwestern University, whose research endeavours to elucidate alterations in neuronal excitability elicited by ALS, SCI and acute-stroke pathophysiology. He received his PhD in Neuroscience, as well as his Master's in Clinical Exercise Physiology, from the University of Illinois at Chicago, and acquired his Bachelor's in Biomedical Sciences from the University of Illinois at Champaign-Urbana. He is a former president of the Shirley Ryan AbilityLab Associate Board, and his personal mission aims to illuminate the mechanistic underpinnings of neural dysfunction in order to develop effectual rehabilitative treatments for differently abled individuals.

D. E. Featherstone is deceased.

**Abstract** In stroke, the sudden deprivation of oxygen to neurons triggers a profuse release of glutamate that induces anoxic depolarization (AD) and leads to rapid cell death. Importantly, the latency of the glutamate-driven AD event largely dictates subsequent tissue damage. Although the contribution of synaptic glutamate during ischaemia is well-studied, the role of tonic (ambient) glutamate has received far less scrutiny. The majority of tonic, non-synaptic glutamate in the brain is governed by the cystine/glutamate antiporter, system $x_c^-$. Employing hippocampal slice electrophysiology, we showed that transgenic mice lacking a functional system $x_c^-$ display longer latencies to AD and altered depolarizing waves compared to wild-type mice after total oxygen deprivation. Experiments which pharmacologically inhibited system $x_c^-$, as well as those manipulating tonic glutamate levels and those antagonizing glutamate receptors, revealed that the antiporter's putative effect on ambient glutamate precipitates the ischaemic cascade. As such, the current study yields novel insight into the pathogenesis of acute stroke and may direct future therapeutic interventions.

(Received 27 September 2022; accepted after revision 18 October 2022; first published online 2 November 2022)

**Corresponding author** J. Larson, Psychiatric Institute (M/C 912), University of Illinois at Chicago, 1601 W. Taylor St., Chicago, IL 60612, USA. Email: jrlarson@uic.edu

**Abstract figure legend** Tonic, extrasynaptic glutamate synergizes with synaptically released glutamate to propagate anoxic depolarization (AD). Mice lacking system $x_c^-$ ($xCT^{-/-}$), an astrocytic glutamate antiporter, exhibit 60–80% lower tonic extracellular glutamate concentrations compared to wild-type controls. Under anoxic challenge, wild-type mice experience excessive presynaptic (Pre) glutamate release, which synergizes with extrasynaptic glutamate to rapidly flood the extracellular space. This activates postsynaptic (Post) AMPA and NMDA receptors, thus precipitating AD, enhancing depolarizing AD waves and provoking the loss of membrane potentials. Despite a greater abundance of postsynaptic AMPA receptors, these ischaemia-driven events are significantly mitigated in $xCT^{-/-}$ mice due to the decrement of tonic glutamate. This ultimately reduces the accumulation of extracellular glutamate during anoxia, delays AD and attenuates depolarizing AD waves in $xCT^{-/-}$ mice. Importantly, both genetic deletion and pharmacological inhibition of system $x_c^-$ produce this ischaemic neuroprotection. Generated using BioRender (www.biorender.com).

## Key points

- Ischaemic stroke remains the leading cause of adult disability in the world, but efforts to reduce stroke severity have been plagued by failed translational attempts to mitigate glutamate excitotoxicity.
- Elucidating the ischaemic cascade, which within minutes leads to irreversible tissue damage induced by anoxic depolarization, must be a principal focus.
- Data presented here show that tonic, extrasynaptic glutamate supplied by system $x_c^-$ synergizes with ischaemia-induced synaptic glutamate release to propagate AD and exacerbate depolarizing waves.
- Exploiting the role of system $x_c^-$ and its obligate release of ambient glutamate could, therefore, be a novel therapeutic direction to attenuate the deleterious effects of acute stroke.

## Introduction

Referring to stroke, Hippocrates commented: 'In violent fits of apoplexy, relief is impossible; in those of a lighter nature, difficult' (Marks, 1817). Unfortunately, more than 2400 years later, little has changed. As the third leading cause of death worldwide and a major contributor to adult disability and lost economic productivity, ischaemic stroke prevails as the most significant neurological disorder in developed and developing countries (Burke et al., 2012). In the USA, someone suffers an ischaemic stroke every 40 s and dies from one every 4 min (Benjamin et al., 2018). Simply stated, stroke remains pernicious due to the unpredictability of its occurrence and the swift time course of ensuing damage.

The human brain accounts for 20–25% of the body's total energy consumption; the largest fraction of ATP is used for maintenance of transmembrane ionic gradients necessary for and perturbed by synaptic and action potentials (Attwell & Laughlin, 2001). In ischaemic stroke, the loss of blood supply to cerebral tissue and the consequent depletion of ATP elicit a

cascade of events leading to the death of neurons and glia within minutes (Lipton, 1999). The rundown of neuronal membrane pumps in the absence of ATP leads to slow depolarization, accelerated action potential firing, synaptic glutamate release, and with further depolarization, release of more glutamate via reversed operation of glutamate transporters (Rossi et al., 2000). This confluence of ischaemia-driven reactions ultimately provokes the loss of membrane potentials, or anoxic depolarization (AD), in affected neurons. Importantly, it has become widely accepted that the AD event is the spark that ignites the neurotoxic sequelae leading to cell death (Ayata, 2018; Hartings et al., 2017; Leão 1947), and the evidence indicates that the timing of AD relative to the onset of anoxia/ischaemia and to the onset of re-oxygenation/reperfusion are critical factors in ischaemic brain damage (Jarvis et al., 2001; Kaminogo et al., 1998; Kostandy, 2012). Shorter AD latencies lead to increased damage (Douglas et al., 2011; Obeidat & Andrew, 1998; Weber & Taylor, 1994) whereas shorter delays between AD and re-oxygenation enhance recovery (Heit et al., 2021). Attenuating the accumulation of excessive extracellular glutamate, which drives AD, is therefore a critical objective for the development of successful therapies. Although prior reports have demonstrated the efficacy of glutamate receptor antagonists in mitigating the ischaemic cascade and delaying AD (Aitken et al., 1988; Fusco et al., 2018; Heit et al., 2021; Roberts et al., 1998), efforts to produce effective interventions for the human condition have been plagued by translational failure due to contraindications resulting from synaptic inhibition of glutamatergic transmission.

The contribution of excitotoxicity and its dependence to acutely released glutamate in ischaemic brain damage is well known (Choi, 1992). The role of ambient glutamate, present tonically in the extracellular space, has received less attention in this regard. Extracellular glutamate is removed after synaptic release by glutamate transporters (primarily excitatory amino acid transporters, EAAT1 and EAAT2), which rapidly import glutamate into glial cells. Tonic, non-synaptic levels of glutamate, however, are regulated primarily by the cystine/glutamate antiporter, system $x_c^-$, which imports cystine from the extracellular compartment and exports glutamate in exchange (Conrad & Sato, 2012): it is estimated that 60–80% of ambient extracellular glutamate arises from the action of this antiporter (Baker et al., 2002; De Bundel et al., 2011; Massie et al., 2011; Ojeda-Torres et al., 2015). Like EAAT1 (Danbolt, 2001), hippocampal system $x_c^-$ is localized to astrocytes (Pow, 2001), which use the imported cystine for glutathione synthesis (Bannai & Tateishi, 1986; Miura et al., 1992). The antiporter therefore influences two principal physiological functions in the brain: antioxidant defence and glutamatergic neurotransmission.

The xCT knockout ($xCT^{-/-}$) mouse with genetic deletion of xCT, the system $x_c^-$ subunit responsible for transport activity, presents an opportunity to test the antiporter's role during ischaemic insult. On one hand, these mice exhibit increased miniature excitatory postsynaptic currents (mEPSCs) and spontaneous excitatory postsynaptic currents (sEPSCs) due to a greater abundance of postsynaptic AMPA receptors (AMPARs) (Williams & Featherstone, 2014), which arguably creates a vulnerability to glutamate excitotoxicity. Furthermore, due to the absence of system $x_c^-$, they lack a vital mechanism for glutathione synthesis *during* oxidative stress (De Bundel, et al., 2011). On the other hand, $xCT^{-/-}$ mice maintain 60–80% lower ambient extracellular glutamate levels (De Bundel et al., 2011; Massie et al., 2011; Ojeda-Torres et al., 2015), which could mitigate ischaemia-driven glutamate excitotoxicity. Thus, $xCT^{-/-}$ mice possess inherent properties which paradoxically promote *both* neuroprotection and/or vulnerability to oxygen deprivation.

The present study therefore examined the contribution of system $x_c^-$ and its regulation of ambient glutamate levels to the timing of anoxic depolarization in an acute *in vitro* model of stroke: hippocampal slices subjected to oxygen deprivation. This model has been extensively used to study the early electrophysiological events in ischaemia (Arrigoni et al., 2005; Coelho et al., 2006; Croning & Haddad, 1998; Dale et al., 2000; Fischer et al., 2009; Heit et al., 2021; Lipton, 1999; Pearson & Frenguelli, 2004; Wang et al., 1999) and replicates the pathophysiological time course of human stroke with fidelity (Dreier et al., 2018). The experiments used wild-type (WT) and $xCT^{-/-}$ mice, as well as a pharmacological inhibitor of system $x_c^-$. We found that mice lacking xCT displayed delayed AD after anoxic challenge, an effect that was reproduced by pharmacological inhibition of system $x_c^-$ in WT mice. Experiments using glutamate receptor antagonists, as well as experiments manipulating tonic glutamate levels, showed that system $x_c^-$ regulates the neuronal response to anoxia via its effects on ambient extracellular glutamate concentrations.

## Methods

### Animals

Experiments were conducted using 2- to 4-month-old male xCT knockout ($xCT^{-/-}$) mice in our colony, bred from mice generously provided by Hideyo Sato (Yamagata University, Japan; Sato et al., 2005) or $xCT^{+/+}$ (WT) controls. The xCT knockout mutation was generated in the C57BL/6J background and backcrossed with C57BL/6J >10 times. xCT expression and function is completely eliminated in these $xCT^{-/-}$ mice. Age-matched C57BL/6J male mice were used as controls. Some experiments were conducted using aged mice

(12–18 months) of both genotypes. All experiments complied with regulations of animal welfare protocols and were approved by the Animal Care Committee at the University of Illinois at Chicago.

Prior studies from our group identified significant differences in AD latency between male and female mice (Figure 10 in Heit et al., 2021); therefore, in order to decrease within-group variance in each experiment, only male cohorts were used for the current investigation. Furthermore, we wanted our results to be comparable with the existent experimental literature elucidating the AD phenomenon, which almost exclusively involves males.

### Electrophysiology

Hippocampal slices were prepared in the conventional manner (Larson & Park, 2009; Larson et al., 1999). Briefly, mice were decapitated and the brain rapidly excised. Anaesthetic was not administered as it can alter subsequent anoxia tolerance in slices (Bickler et al., 2005). The hippocampus was then dissected free, and slices cut on a tissue chopper at 400 $\mu$m transverse to the long axis. Slices were then maintained in an interface chamber at 34°C, and continually perfused with artificial cerebrospinal fluid (ACSF) containing the following (in mM): NaCl 124, KCl 3.0, $KH_2PO_4$ 1.0, $NaHCO_3$ 26, $MgSO_4$ 2.0, $CaCl_2$ 2.0, D-glucose 10, and sodium L-ascorbate 1.9 (unless specified otherwise), gassed with 95% $O_2$ and 5% $CO_2$. No slices were exposed to more than one bout of anoxia. This is important because anoxia exposure may inflict damage or produce a 'preconditioning' effect (Kirino, 2002). Recordings were made in the interface chamber with constant perfusion (1.0 ml min$^{-1}$) of ACSF at 34°C with the upper surface exposed to an atmosphere of 95% $O_2$ and 5% $CO_2$. The gas supplied to the chamber flowed at a rate of 1 litre min$^{-1}$.

Two stimulating electrodes were utilized to evoke field potentials, which were recorded with glass electrodes filled with 2 M NaCl (1–5 MΩ). To monitor synaptic transmission, a stimulating electrode was placed in stratum radiatum of field CA1c to activate Schaffer-commissural (SC) fibre-evoked synaptic field potentials recorded in stratum radiatum of CA1b. Antidromic spikes were recorded in stratum pyramidale in response to stimulation of the alveus. Laminar profiles were used to place electrodes optimally in each slice tested. In paired experiments, where two slices were compared within the same experimental conditions, only one response (either orthodromic or antidromic) was taken from each slice. We also continuously monitored all recordings at low frequency on an oscilloscope. All slices from each animal were tested before experiments, and one or two slices exhibiting the largest field potentials were selected for study.

Evoked responses were amplified (100–500×), filtered (bandpass 0.1–5 kHz), digitized by microcomputer (PC), and analysed on-line using custom software (Labview, National Instruments, Austin, TX, USA). Excitatory postsynaptic potentials (fEPSPs) or antidromic spikes were evoked at 10-s intervals throughout the experiments. Baseline stimulus intensity was set to evoke a half-maximal fEPSP in each slice. Baseline recordings were taken for at least 12 min prior to manipulations. Initial slope and peak amplitude were calculated for each fEPSP and normalized to the baseline average in each slice. Antidromic spikes were quantified as the peak amplitude of the negative spike from the post-stimulus baseline. In some experiments, paired pulse facilitation (PPF) was assessed using stimuli separated by intervals ranging from 50 to 800 ms. In these cases, PPF was calculated as the percentage increase in the amplitude of the second response relative to that of the first response of the pair.

Specific parameters were implemented in determining which hippocampal slices would be used for experimentation. Slices deemed acceptable for measurements had to display (i) the capacity to generate an evoked potential of at least 4.0 mV in amplitude, (ii) fEPSPs which maintained a half-width ≤7.0 ms for the duration of the baseline period, and (iii) PPF values between 135% and 165% during the duration of the baseline period. Furthermore, throughout the baseline period, amplitudes of individual responses did not deviate more than ±0.2 mV from the initial response captured at the start of the baseline period. This was to verify that evoked responses were stable and not 'trending' upward or downward prior to anoxic intervention or drug perfusion. Any slices which did not meet these criteria were excluded from testing.

### Anoxia

Anoxia was induced by $N_2$ totally replacing $O_2$. We refer to this condition as 'nominal anoxia' as the slice chamber is an open system, potentially contaminated with trace $O_2$ from the outside air. In these experiments, nominal anoxia was maintained until slices displayed anoxic depolarization (AD), signalled by the loss of synaptic response, a large slow direct current (DC) shift in the field potential (the 'AD wave'), and abolition of the presynaptic fibre volley and antidromic spike. The AD wave was observed using an oscilloscope and its latency from anoxia onset (i.e. 'AD latency') was recorded with a stopwatch. In all anoxia experiments, the field potential (filtered bandpass 0.1–50 Hz) was continuously recorded by computer (digitized at 100 Hz) to display AD waves in figures along with evoked responses. The AD wave onset was taken to be the time when the first digitized point

had a negative voltage greater than $-0.25$ mV and the AD wave end was taken to be the time point after the AD peak when the negative wave reached $-0.25$ mV. The 'AD wave amplitude' was measured as the peak negativity while the 'AD wave duration' was measured as the difference between AD wave onset and end (see Fig. 2*B*). Slices were re-oxygenated (replacement of $N_2$ with $O_2$) 60 s after AD and recordings continued for 45 min to examine extent of recovery of evoked field potentials in certain experiments. In conditions where synaptic transmission was pharmacologically antagonized, antidromic responses were monitored and used to record AD latency. The epoch from the onset of oxygen deprivation to the time of re-oxygenation is referred to as the 'anoxic period'. The epoch from the onset of oxygen deprivation to a 25% decline in synaptic transmission was also measured in certain experiments and is referred to as the 'anoxic response (AR) onset'.

### Pharmacology

All drugs were obtained from Tocris Bioscience/ Bio-Techne (Minneapolis, MN, USA). The AMPA receptor antagonist 6-cyano-7-nitroquinoxaline-2,3-dione (CNQX) was dissolved in 100% dimethyl sulfoxide (DMSO) at 50 mM and diluted in ACSF to a final concentration of 50 $\mu$M. The NMDA receptor antagonist D-2-amino-5-phosphonopentanoic acid (AP5) was dissolved in water at 50 mM and diluted in ACSF to a final concentration of 50 or 250 $\mu$M. The mGluR5 antagonist 2-methyl-6-(phenylethynyl)pyridine (MPEP) was dissolved in 100% DMSO at 50 mM and diluted in ACSF to a final concentration of 50 $\mu$M. System $x_c^-$ inhibitor *S*-4-carboxyphenylglycine (CPG) was dissolved in NaOH (0.1 M) at 50 mM and diluted in ACSF to a final concentration of 100 $\mu$M. 'Vehicle' treatment used the same solution without the added drug.

### Experimental design and statistical analyses

Data are presented as means $\pm$ standard deviation (SD) (*n* is number of animals in the sample). Statistical analyses were performed using Prism 9 (GraphPad Software, San Diego, CA, USA). For comparisons involving one independent variable and two groups, Student *t* test (two-tailed) was used. Slices treated in the same chamber at the same time were treated as paired samples. Mean differences were considered significant at $P < 0.05$. Comparisons involving more than one independent variable utilized the two-way analysis of variance (ANOVA). When ANOVA was significant, planned comparisons were conducted using Šidák's test.

## Results

### xCT$^{-/-}$ mice display protracted AD latency, attenuated AD wave amplitude and increased AD wave duration, but do not differ in paired-pulse facilitation

Our first objective was to determine whether system $x_c^-$ participates in the ischaemic cascade. We modelled ischaemia by subjecting hippocampal slices to total oxygen deprivation. In the interface chamber, the upper surfaces of slices are exposed to an atmosphere of carbogen gas (95% $O_2$, 5% $CO_2$) with the submerged surfaces continually perfused with ACSF. AMPA receptor (AMPAR)-mediated synaptic responses evoked by Schaffer-commissural fibre stimulation in CA1 were monitored throughout the experiments. To induce anoxia, the $O_2$ in the carbogen mixture was replaced with $N_2$. In each experiment, nominal anoxia was maintained until slices displayed anoxic depolarization (AD), signalled by (i) the loss of synaptic response, (ii) a large, slow shift in the field potential ('the AD wave'), and (iii) abolition of the presynaptic fibre volley and antidromic response, as previously described (Heit et al., 2021). These events appear simultaneously after oxygen deprivation, and the elapsed time required for their appearance was recorded as the AD latency for each slice.

For our initial investigation, we performed paired experiments where hippocampal slices from groups of 2- to 4-month-old WT ( xCT$^{+/+}$) and xCT-knockout (xCT$^{-/-}$) mice were exposed to anoxia within the same chamber at the same time. Figure 1*A* and *B* displays a representative experiment comparing WT and xCT$^{-/-}$ mice, where the synaptic response (fEPSP) was monitored for each genotype, and Fig. 1*D* displays a schematic representation of the placement of stimulating and recording electrodes. Figure 2*A* shows the evoked synaptic responses overlain at higher temporal resolution for the anoxic period of the same representative experiment. Synaptic responses in the WT slice began to decrease 10–30 s prior to the appearance of the corresponding AD wave (Fig. 2*B*) and the simultaneous abolition of the synaptic response and the fibre volley (AD latency = 108 s). The xCT$^{-/-}$ slice showed a qualitatively similar response to anoxia; however, the AD was delayed by 2 min (AD latency = 229 s; 112% longer than WT). Data from 14 paired experiments showed a robust difference in AD latency between WT and xCT$^{-/-}$ slices (paired $t_{13} = 4.772$, $P < 0.0004$, Fig. 2*C*). Interestingly, the anoxic response onset (AR onset), defined as the time required for anoxia to reduce the synaptic response by $\geq 25\%$, did not differ between genotypes (paired $t_{13} = 1.906$, $P = 0.0789$, Fig. 2*F*). These results indicate that the earliest detectable electrophysiological response to anoxia is similar in slices from WT and

xCT$^{-/-}$ mice; however, the AD is significantly delayed in xCT$^{-/-}$ slices.

We also measured the amplitude and duration of the depolarizing waves ('AD waves') for each slice at the time of anoxic depolarization (Fig. 2*B*). Like other slow field potentials, the magnitude and time course of the AD wave depends on (i) the number of participating neurons, (ii) the magnitude of response of each neuron, (iii) the rate of change in the response in individual neurons, and (iv) the degree of synchrony of responses among the population of neurons in the recording field (Heit et al., 2021). AD waves in xCT$^{-/-}$ mice were flattened compared to WT mice, with attenuated amplitudes (paired $t_{13} = 3.625$,

$P = 0.0031$, Fig. 2*D*) and increased durations (paired $t_{13} = 2.335$, $P = 0.0363$, Fig. 2*E*). This would be expected if the longer AD latencies in xCT$^{-/-}$ mice were associated with reduced synchrony in different neurons, although we cannot rule out slower depolarization in single neurons. A Pearson's correlation coefficient was also computed to assess the linear relationship between AD wave amplitude and AD latency for both genotypes. Interestingly, WT slices displayed a negative correlation ($r = -0.6557$, $P = 0.0109$, data not shown) between the two variables, whereas xCT$^{-/-}$ slices showed no correlation ($r = 0.2258$, $P = 0.4377$, data not shown). This difference between genotypes in these relationships

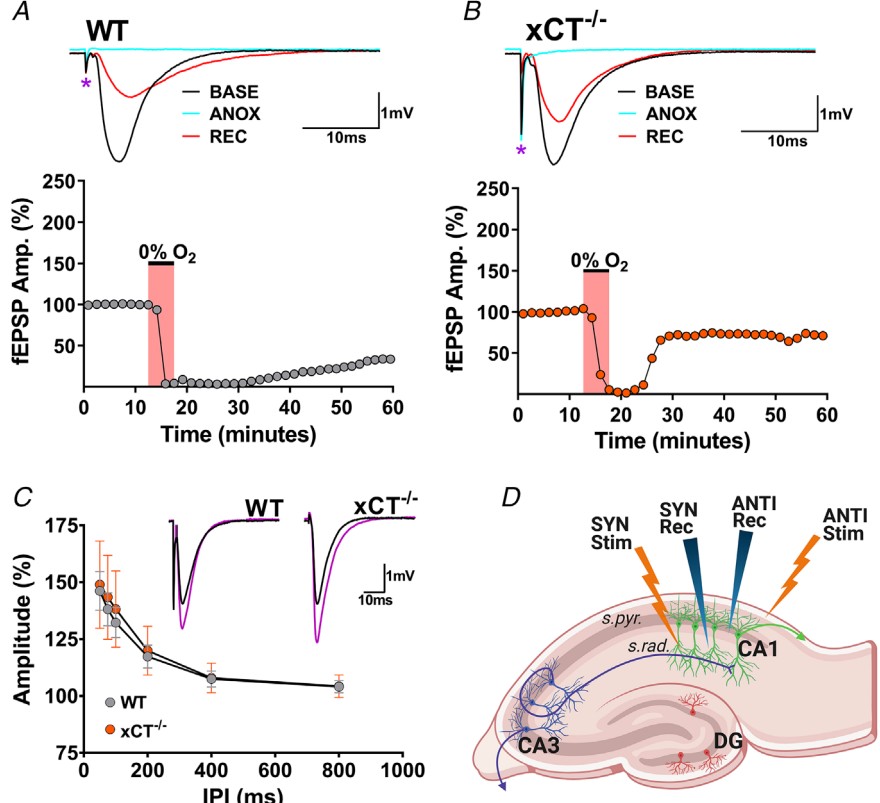

**Figure 1. Design for paired anoxia experiments comparing WT and xCT$^{-/-}$ slices**

*A* and *B*, waveform traces (top) show averaged ($n = 5$) individual responses for WT (*A*) and xCT$^{-/-}$ (*B*) slices in the same chamber during baseline (BASE, black; immediately prior to anoxia), anoxia (ANOX, blue; last trial before re-oxygenation), and recovery (REC, red; 45 min after re-oxygenation) periods. Magenta asterisk denotes the stimulus artifact. Graphs (bottom) show normalized measurements of synaptic responses (fEPSP Amp.) from WT (*A*) and xCT$^{-/-}$ (*B*) slices in the same chamber before, during and after an episode of anoxia. Each circle represents the average amplitude of five consecutive responses normalized to the pre-anoxic baseline average for WT (grey) and xCT$^{-/-}$ (orange) slices. WT and xCT$^{-/-}$ responses were evoked alternately at 10 s intervals. The pink area denotes the anoxic period, which was terminated 60 s after the appearance of the AD wave in the xCT$^{-/-}$ slice (see text). *C*, paired-pulse facilitation (PPF) graph showing excitatory postsynaptic potential (fEPSP) initial amplitude for second response as a percentage of the first response of the pair with inter-pulse intervals (IPIs) of 50, 75, 100, 200, 400 and 800 ms. Slices from both WT (grey circles) and xCT$^{-/-}$ (orange circles) slices show robust facilitation at IPIs of 200 ms or less with marginal facilitation at 400 and 800 ms. Each data point represents the average of 10 responses from 10 different slices (means ± SD) at a given IPI. Insets show representative individual traces (black) for WT and xCT$^{-/-}$ slices with attendant potentiated responses (magenta) at the 75 ms IPI. *D*, schematic representation of synaptic organization in the hippocampal slice and placement of synaptic (SYN) and antidromic (ANTI) stimulating and recording electrodes. Generated using BioRender (www.biorender.com).

[Colour figure can be viewed at wileyonlinelibrary.com]

suggests that anoxia-induced depolarizing events in mutant slices are desynchronized due to the decrement of ambient glutamate. Additionally, we compared the magnitude of paired-pulse facilitation (PPF) between genotypes (prior to anoxia) using inter-pulse intervals (IPIs) of 50, 75, 100, 200, 400 and 800 ms. Our findings showed no difference in PPF for xCT$^{-/-}$ slices when compared to control slices at any IPI (Fig. 1*C*). This indicates that the xCT mutation creates no detectable effect on the presynaptic neurotransmitter release mechanisms which result in short-term plasticity.

Mouse brain expression of xCT protein increases throughout development, peaks in adulthood (La Bella et al., 2007), and is believed to be maintained throughout mid- and late life (Shih et al., 2006). The altered AD characteristics seen in young mice were reproduced in comparisons of WT and xCT$^{-/-}$ slices obtained from mice aged 12–18 months. Aged xCT$^{-/-}$ mice displayed longer AD latencies (paired $t_6 = 9.871$, $P = 0.0001$, Fig. 2*C*), as well as decreased AD wave amplitudes (paired $t_6 = 3.019$, $P = 0.0234$, Fig. 2*D*) and longer AD wave durations (paired $t_6 = 2.773$, $P = 0.0323$, Fig. 2*E*) when compared to age-matched WT controls.

### xCT$^{-/-}$ and WT slices show similar recovery of synaptic transmission with re-oxygenation 1 min after AD

In the paired experiments illustrated in Figs 1 and 2, WT and xCT$^{-/-}$ slices from young mice were re-oxygenated 60 s after whichever slice displayed the longer AD latency, and then allowed to recover for 45 min. Because WT slices reached AD earlier, xCT$^{-/-}$ slices remained in the depolarized state for shorter epochs relative to WT slices. Importantly, the time between AD and re-oxygenation strongly influences recovery (Heit et al., 2021); hence, it is not surprising that xCT$^{-/-}$ slices showed improved recovery of fEPSPs (55.640 ± 21.950% of pre-anoxia baseline) compared to WT slices

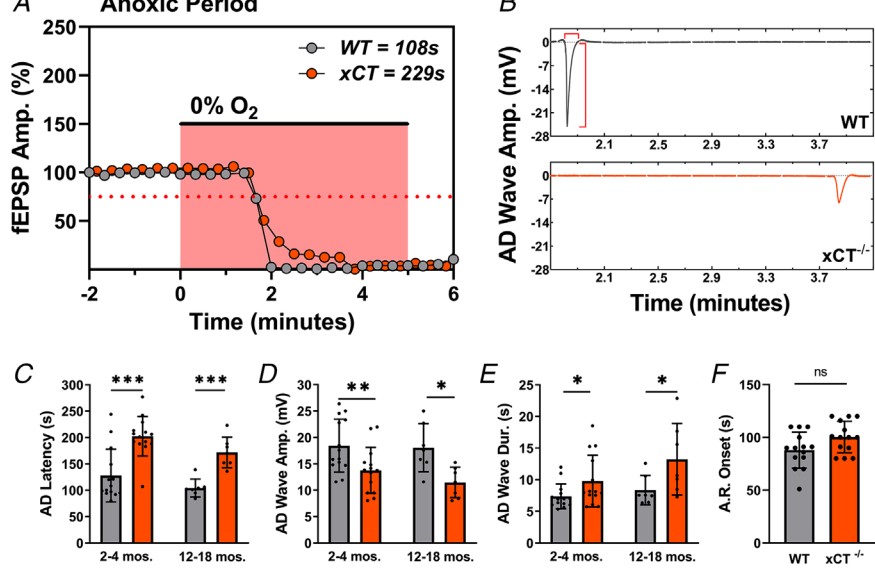

**Figure 2. xCT$^{-/-}$ mice display enhanced anoxia tolerance compared to WT controls**
*A* and *B*, data displayed from same representative experiment as in Fig. 1*A* and *B*. *A*, the expanded anoxic period graph illustrating changes in WT and xCT$^{-/-}$ responses with greater temporal resolution. The abscissa shows time relative to anoxia onset. Each point represents a single evoked response for WT (grey) and xCT$^{-/-}$ (orange) synaptic responses. For this paired experiment, the WT slice recorded an AD latency of 108 s, and the xCT$^{-/-}$ slice recorded an AD latency of 229 s. Horizontal red dotted line highlights 75% of baseline fEPSPs, which was the threshold used to calculate anoxic response onset in *F*. Note that both responses begin to decrease at the same time point, but the xCT$^{-/-}$ response is maintained about 2 min longer than the WT. *B*, segments of continuous recordings from CA1 stratum radiatum (s.rad) in WT and xCT$^{-/-}$ slice showing large, negative potential ('AD wave'). The elimination of the fibre volley and synaptic response was coincident with the appearances of AD waves for both the WT and xCT$^{-/-}$ slice. Vertical red bracket highlights the measurement of the AD wave amplitude, and the horizontal red bracket denotes time period used to measure AD wave duration. Time axis relative to anoxia onset as in *A*. *C–E*, bar graphs showing paired analyses of young (2 to 4 months) and aged (12–18 months) WT and xCT$^{-/-}$ slices for AD latency, AD wave amplitude and AD wave duration. *F*, bar graph showing paired analysis of anoxic response onset (AR onset) measured as the duration of time required for fEPSPs to reach 75% of baseline in young WT and xCT$^{-/-}$ slices. (paired student's *t* test, *$P < 0.05$, **$P < 0.01$, ***$P < 0.001$; ns, not significant). [Colour figure can be viewed at wileyonlinelibrary.com]

(27.190 ± 19.86%; paired $t_{13} = 3.600$, $P = 0.0032$) in these paired experiments.

We therefore sought to determine if recovery of synaptic transmission would differ between genotypes when the time between AD and re-oxygenation was held constant. As such, we performed anoxia experiments where xCT$^{-/-}$ or WT slices were individually subjected to anoxia, measured for latency to anoxic depolarization, left in the depolarized state for 60 s, then fully re-oxygenated and allowed to recover for 45 min. In addition to synaptic transmission, we also monitored the antidromic response of CA1 neurons to direct stimulation of their axons. Unlike the synaptic response, which can be influenced by endogenous neuromodulators such as adenosine, the antidromic response is maintained until the abrupt loss of membrane potentials that occurs with AD (Heit et al., 2021). At the onset of anoxia, ATP breakdown releases adenosine into the extracellular space, which binds to pre-synaptic receptors, thus causing the synaptic response to gradually decline (Dunwiddie & Masino, 2001). The antidromic response, however, is maintained until the loss of membrane potentials, and is therefore time-locked to AD (Heit et al., 2021).

Figure 3 displays representative unpaired experiments for both WT and xCT$^{-/-}$ mice. Mutant mice exhibited a significant increase in AD latency for both the synaptic ($t_{12} = 5.515$, $P < 0.0001$, Fig. 3C) and antidromic ($t_{12} = 5.489$, $P < 0.0001$, Fig. 3F) response when compared to WT counterparts. In agreement with prior findings, the AD wave in the dendritic layer (s. radiatum) always preceded the AD wave in the somatic layer (s. pyramidale) for WT slices (mean difference: 1.857 ± 0.690 s, paired $t_6 = 7.120$, $P = 0.0004$), confirming that the depolarizing wave originates in the dendritic layer and propagates to the somatic layer (Heit et al., 2021). In mutant slices, however, the appearance of the AD waves for each region were closer together in time, and the somatic AD wave appeared first in two experiments (mean difference: 0.143 ± 1.069 s, paired $t_6 = 0.355$, $P = 0.736$). The magnitude of post-anoxic recovery of the synaptic response, measured as a percentage of the baseline response, did not differ between genotypes ($t_{12} = 0.6401$, $P = 0.5341$, Fig. 3I). Notwithstanding, we cannot rule out the possibility that an extended re-oxygenation period (i.e. >45 min) would have revealed a differential effect. As can be seen in the figure, recovery could be quite variable: in most experiments the range was between 50% and 100% of the baseline responses with one case in each data set measuring below that range. Lastly, the magnitude of PPF after the recovery phase did not differ between WT and xCT$^{-/-}$ slices ($t_{12} = 0.4222$, $P = 0.6803$; data not shown) suggesting that xCT mutation does not influence alterations in this short-term plasticity mechanism during the acute phase of ischaemic recovery.

## Restitution of ambient glutamate in xCT$^{-/-}$ slices normalizes AD latency

Microdialysis studies show that extracellular glutamate levels *in vivo* in hippocampus and striatum are 60% lower in xCT$^{-/-}$ mice compared to WT mice (De Bundel et al., 2011; Massie et al., 2011); these results were replicated in hippocampal slices using low-flow push–pull perfusion methods (Ojeda-Torres et al., 2015). Lower ambient glutamate could be a crucial variable in the delayed rate of depolarization observed in xCT$^{-/-}$ mice during anoxic conditions. To test this, hippocampal slices from WT and xCT$^{-/-}$ mice were co-incubated in ACSF containing 5 μM added glutamate, subjected to anoxia, and measured for latency to anoxic depolarization. The added glutamate mimics extracellular glutamate concentrations (5 μM) observed in WT hippocampal slices (Ojeda-Torres et al., 2015). Figure 4A–G summarizes the results from paired anoxia experiments performed with 5 μM glutamate ACSF. The addition of glutamate to the ACSF bath eliminated the difference in AD latency between WT and xCT$^{-/-}$ slices (paired $t_7 = 0.231$, $P = 0.824$, Fig. 4D). Concomitantly, AD wave amplitudes (paired $t_7 = 0.325$, $P = 0.755$, Fig. 4E) and AD wave durations (paired $t_7 = 0.0132$, $P = 0.990$, Fig. 4F) showed no significant differences between genotypes. The onset of the anoxic response also did not differ between genotypes (paired $t_7 = 0.664$, $P = 0.528$, Fig. 4G). Further statistical analyses, which compared these data with results from age-matched animals in control conditions (glutamate-free ACSF, Fig. 2C), yielded significant main effects for genotype ($F_{1,40} = 9.66$, $P = 0.0035$, 2-way ANOVA) and for ACSF glutamate concentration ($F_{1,40} = 18.50$, $P = 0.0001$) on the latency to AD. The interaction between genotype and ACSF glutamate concentration was also significant ($F_{1,40} = 10.76$, $P = 0.0022$). Comparisons between AD latencies in glutamate-free ACSF conditions (Fig. 2C) *versus* 5 μM glutamate ACSF conditions (Fig. 4D) were significant for xCT$^{-/-}$ slices ($P < 0.0001$), but not for WT slices ($P = 0.9790$) (Šidák's test for multiple comparisons).

## WT mice mimic xCT$^{-/-}$ mice AD latencies with wash-out of ambient glutamate

Since the addition of ambient glutamate to the ACSF bath caused xCT$^{-/-}$ slices to phenocopy WT slices, we next tested whether the opposite would be true: would the wash-out of extracellular hippocampal glutamate cause WT mice to phenocopy xCT$^{-/-}$ mice? Previous studies have revealed that prolonged incubation periods (≥5 h) in (glutamate-free) ACSF result in a significant decrease of extracellular glutamate levels in WT slices, while levels in xCT$^{-/-}$ slices remain unchanged (Cabay et al., 2018; Ojeda-Torres et al., 2015). It therefore followed that we should test whether

prolonged incubation in glutamate-free ACSF would extend AD latency in WT slices but not in xCT$^{-/-}$ slices. Hippocampal slices from groups of WT and xCT$^{-/-}$ mice were incubated for 6–7 h at a perfusion rate of 1 ml min$^{-1}$, subjected to anoxia, and measured for latency to anoxic depolarization. Figure 4H–N summarizes the

results from paired glutamate wash-out experiments. Extended incubation conditions eliminated the difference in AD latency between WT and xCT$^{-/-}$ slices (paired $t_{10} = 0.350$, $P = 0.734$, Fig. 4K). Concomitantly, AD wave amplitudes (paired $t_9 = 0.640$, $P = 0.538$, Fig. 4L) and AD wave durations (paired $t_9 = 1.576$, $P = 0.1494$, Fig. 4M)

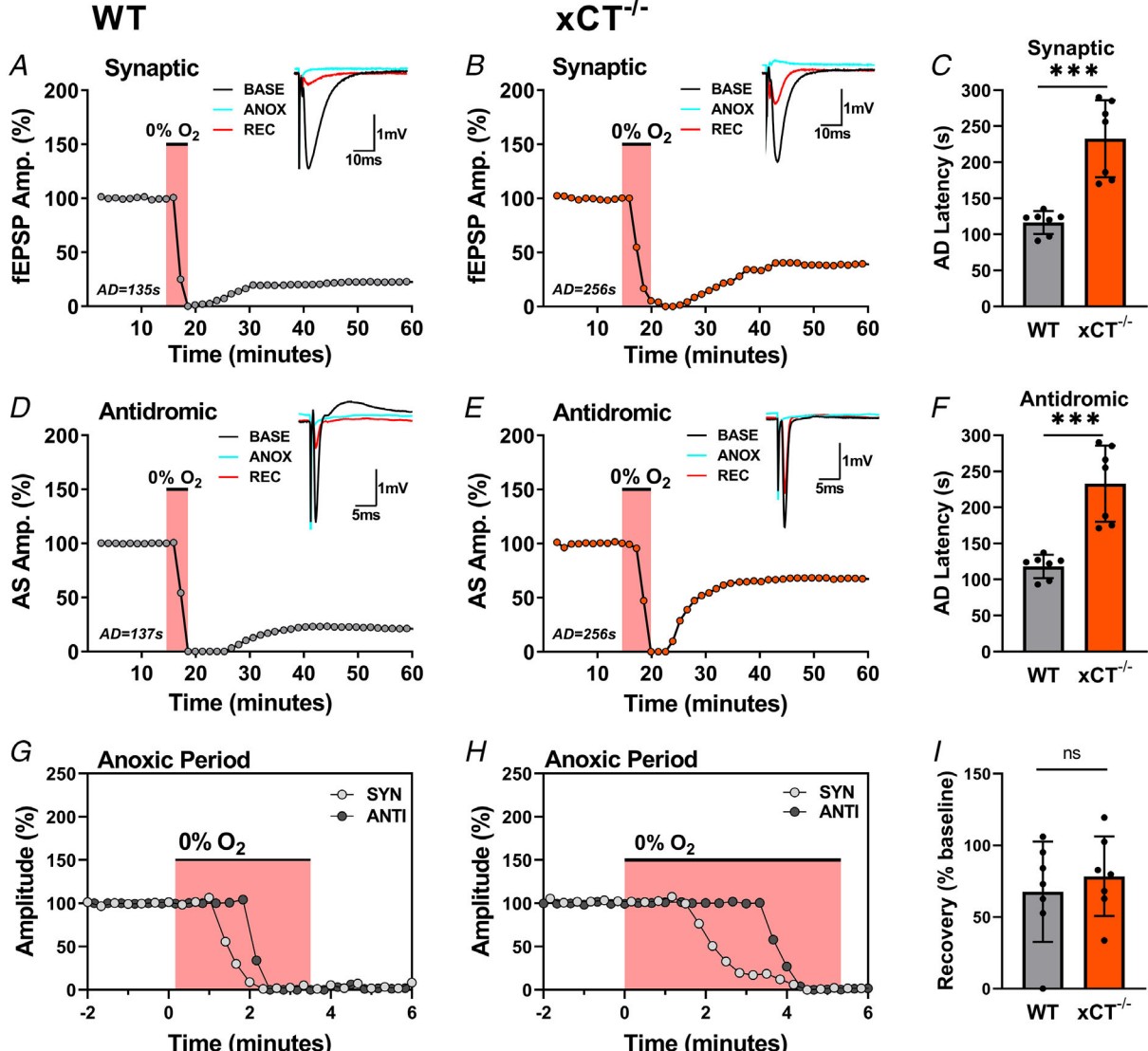

**Figure 3. xCT$^{-/-}$ slices show similar recovery to WT after 60 s in depolarized state**
*A* and *B*, measurement of synaptic responses (fEPSP Amp.) in s. radiatum of WT and xCT$^{-/-}$ slices shown before, during and after an episode of anoxia (pink area). Each point represents the average of four trials and insets show averaged ($n = 4$) individual responses during the baseline (black), anoxic (blue), and recovery (red) periods. The AD latencies recorded in the dendritic field of these WT and xCT$^{-/-}$ slices were 135 and 256 s, respectively. *C*, bar graph comparing AD latencies recorded in s. radiatum for unpaired WT and xCT$^{-/-}$ slices. *D* and *E*, measurement for antidromic spikes (AS Amp.) in the same two experiments. The AD latencies recorded from s. pyramidale in these WT and xCT$^{-/-}$ slices were 137 and 256 s, respectively. *F*, bar graph comparing AD latencies recorded in s. pyramidale for unpaired WT and xCT$^{-/-}$ slices. *G* and *H*, superimposed measurements of single-trial responses at expanded time scale (anoxia onset at t = 0) comparing synaptic and antidromic responses from the same WT (*G*) and xCT$^{-/-}$ (*H*) slices. *I*, bar graph comparing magnitude of recovery 45 min after re-oxygenation measured as percentage of baseline fEPSPs for WT (67.67 ± 35.01) and xCT$^{-/-}$ (78.49 ± 27.79). Slices remained in depolarized state for 60 s after AD prior to re-oxygenation. See schematic representation in Fig. 1D for placement of synaptic (s. rad) and antidromic (s. pyr) stimulating and recording electrodes. ***$P < 0.001$; ns, not significant. [Colour figure can be viewed at wileyonlinelibrary.com]

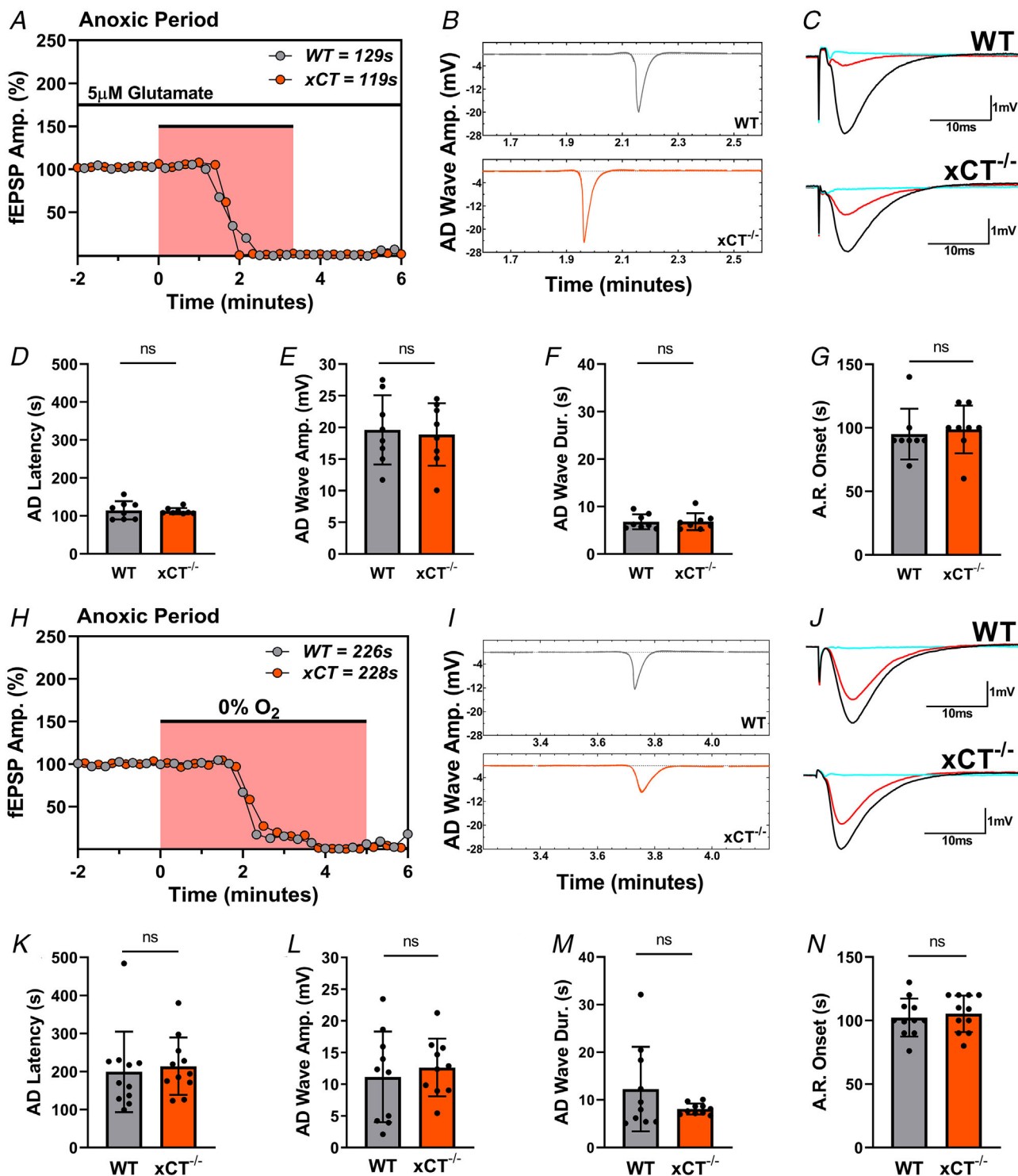

**Figure 4. Restitution and wash-out of ambient glutamate eliminate difference in anoxia tolerance between WT and xCT$^{-/-}$ slices**

*A–G*, results from paired anoxia experiments, where WT and xCT$^{-/-}$ slices were co-incubated in 5 $\mu$M glutamate ACSF. *A*, representative anoxic period graph highlighting changes in WT (grey) and xCT$^{-/-}$ (orange) synaptic responses, where each point represents a single evoked response. The abscissa shows time relative to anoxia. For this paired experiment, the WT slice recorded an AD latency of 129 s, and the xCT$^{-/-}$ slice recorded an AD latency of 119 s. *B*, the elimination of the fibre volley and synaptic response was coincident with the appearances of AD waves for both the WT and xCT$^{-/-}$ slice. Paired slices showed near simultaneous appearance of AD waves

in 5 $\mu$M glutamate ACSF. Time axis relative to anoxia onset as in *A*. *C*, waveform traces showing averaged ($n = 5$) individual responses during baseline (black), anoxic (blue) and recovery (red) periods for WT and xCT$^{-/-}$ slice. *D–G*, bar graphs showing paired analyses of WT and xCT$^{-/-}$ slices for AD latency, AD wave amplitude, AD wave duration and anoxic response onset. *H–N*, results from paired anoxia experiments in glutamate-free ACSF after prolonged incubation of 6–7 h. *H*, representative anoxic period graph highlighting changes in WT (grey) and xCT$^{-/-}$ (orange) synaptic responses, where each point represents a single evoked response. The abscissa shows time relative to anoxia. For this paired experiment, the WT slice recorded an AD latency of 226 s, and the xCT$^{-/-}$ slice recorded an AD latency of 228 s. *I*, the elimination of the fibre volley and synaptic response was coincident with the appearances of AD waves for both the WT and the xCT$^{-/-}$ slice. Paired slices showed near simultaneous appearance of AD waves after prolonged incubation duration. Time axis relative to anoxia onset as in *H*. *J*, waveform traces showing averaged ($n = 5$) individual responses during baseline (black), anoxic (blue) and recovery (red) periods for WT and xCT$^{-/-}$ slice. *K–N*, bar graphs showing paired analyses of WT and xCT$^{-/-}$ slices for AD latency, AD wave amplitude, AD wave duration and anoxic response onset (AD wave amplitude and duration could not be measured in one experiment). Both the 5 $\mu$M glutamate-ACSF and the glutamate wash-out conditions eliminated the difference in anoxia responses between genotypes. ns, not significant. [Colour figure can be viewed at wileyonlinelibrary.com]

showed no significant differences between genotypes after extended incubation, nor was there a difference in anoxic response onset (paired $t_{10} = 0.625$, $P = 0.535$ Fig. 4*N*). From the 11 paired wash-out experiments, five WT slices and six xCT$^{-/-}$ slices displayed the longer AD latency out of the pair, and were thus left in the depolarized state for exactly 1 min before reoxygenation. This allowed us to empirically measure and compare post-anoxic recovery between genotypes. No differential effect in recovery, however, was observed after extended incubation ($t_9 = 0.2035$, $P = 0.8433$; data not shown).

Further statistical analyses, which compared wash-out data with those from short incubation durations (2–3 h, Fig. 2*C*), yielded significant main effects for genotype ($F_{1,46} = 5.579$, $P = 0.0225$, 2-way ANOVA) and incubation duration ($F_{1,46} = 4.235$, $P = 0.0453$) on the latency to AD. The interaction effect between genotype and incubation duration, however, failed to reach significance ($F_{1,46} = 2.053$, $P = 0.1586$). Comparisons between AD latencies in short (2–3 h, Fig. 2*C*) *versus* extended (6–7 h, Fig. 4*K*) incubation conditions were significant for WT slices ($P = 0.0344$), but not for xCT$^{-/-}$ slices ($P = 0.8848$) (Šidák's test for multiple comparisons). Thus, extended perfusion with glutamate-free ACSF delayed AD in WT slices but not xCT$^{-/-}$ slices.

Concerning the data in Fig. 4, it is worth noting that the expression of glutamate transporters (EAATs) is not completed until P20 in murine subjects (Kugler & Schleyer, 2004), and levels of total glutamate concentration can double between P14 and P21 in the hippocampus (Tkac et al., 2003). As such, our experiments used only adult mice between 2 and 4 months of age in order to control for these developmental changes. Furthermore, a perfused incubation chamber is a requisite component of slice electrophysiology, but this is the first known observation indicating that incubation time can influence ischaemic sensitivity. Future investigations should closely monitor perfusion rate and incubation duration, and, if possible, any fluctuations in amino acid concentrations within tissue samples.

## Glutamate receptor antagonism increases AD latency and eliminates differences between WT and xCT$^{-/-}$ mice

The results presented thus far suggest that system x$_c^-$ influences the timing of AD by regulating the tonic extracellular concentrations of glutamate. Since glutamate receptors are known to be involved in the occurrence and timing of AD (Heit et al., 2021), it might be expected that blockade of glutamate receptors would eliminate the differences in AD latency between WT and xCT$^{-/-}$ slices. Accordingly, for 40 min prior to anoxia and measurement of AD latency, slices from WT and xCT$^{-/-}$ mice were incubated with a cocktail of glutamate receptor antagonists: (i) 50 $\mu$M MPEP, a selective metabotropic glutamate receptor 5 (mGluR5) antagonist, (ii) 50 $\mu$M CNQX, a competitive AMPA/kainate glutamate receptor antagonist, and (iii) 50 $\mu$M D-AP5, a selective NMDA receptor (NMDAR) antagonist. Due to the inhibition of AMPAR-driven synaptic transmission by CNQX, anti-dromic responses were recorded for both WT and xCT$^{-/-}$ slices throughout the experiments (Fig. 5). To our surprise, both groups showed a modest increase in AD latency after treatment with the antagonists; xCT$^{-/-}$ latencies remained significantly higher than those of WT slices (paired $t_{11} = 7.912$, $P < 0.0001$, Fig. 5*E*). Additionally, AD wave amplitudes remained decreased (paired $t_{11} = 2.313$, $P = 0.041$, Fig. 5*F*) and AD wave durations remained lengthened (paired $t_{11} = 2.236$, $P = 0.047$, Fig. 5*G*) in xCT$^{-/-}$ slices when compared to WT.

Although the concentrations of the competitive antagonists used were sufficient to abolish synaptic trans-mission prior to anoxia, it is possible that the massive release of glutamate might overwhelm the antagonists during AD. Because NMDARs are the primary drivers in glutamate excitotoxicity (Choi, 1992; Dirnagl et al., 1999), we repeated the experiments with a five-fold increase in dosage for NMDA antagonist, D-AP5 (Fig 6). As such, slices from both groups were subjected to anoxia after being pretreated and continuously perfused with

(i) 50 $\mu$M MPEP, (ii) 50 $\mu$M CNQX, and (iii) 250 $\mu$M D-AP5. With higher concentrations of D-AP5, the AD latencies of WT slices matched those of xCT$^{-/-}$ slices (paired $t_8 = 0.005$, $P = 0.9965$, Fig. 6$E$). Additionally, there were no differences in AD wave amplitudes (paired $t_6 = 0.296$, $P = 0.7775$, Fig. 6$F$) or AD wave durations (paired $t_6 = 1.139$, $P = 0.2981$, Fig. 6$G$) between genotypes in conditions of higher D-AP5 dosage.

### Pharmacological inhibition of system x$_c^-$ in WT slices reproduces effect of xCT KO on AD

We next aimed to substantiate the conclusion that xCT$^{-/-}$ mice differ from WT mice due to the absence of the antiporter and not a compensatory change in other genes. As such, we applied *S*-4-carboxyphenylglycine (CPG, 100 $\mu$M), a non-substrate inhibitor of the antiporter (Patel et al., 2004), to WT and xCT$^{-/-}$ slices prior to the anoxic episode. Importantly, CPG was chosen over sulfasalazine (SAS) in this regard, because SAS (Ryu et al., 2003) and derivatives of SAS (Cho et al., 2010; Gwag et al., 2007) have been shown to inhibit NMDARs in models of ischaemia.

Again, the AD event is predominantly precipitated by the activation of NMDARs (Fusco et al., 2018; Heit et al., 2021), and therefore CPG proved to be the best option for this investigation.

We first ran separate paired experiments to assess whether 2.0 h of CPG administration would affect normal synaptic transmission in WT and/or xCT$^{-/-}$ slices. (The 2-h perfusion epoch was chosen because, unlike the antagonism of synaptic transmission via CNQX, CPG must inhibit xCT-mediated glutamate release for a sufficient duration in order for ambient levels to decrease.) Figure 7 highlights the effects of 2.0 h of CPG versus vehicle treatment for WT (Fig. 7$A$–$C$) and xCT$^{-/-}$ (Fig. 7$D$–$F$) slices during normoxic conditions. In both genotypes, CPG treatment elicited a gradual increase in fEPSPs, which stabilized after 90 min. A two-way ANOVA was performed with paired slices run as repeated measures for the genotype variable (WT *vs.* xCT$^{-/-}$) and drug treatment (CPG *vs.* vehicle) run as the between-subjects variable. This yielded a significant main effect for drug treatment ($F_{1,18} = 23.70$, $P = 0.0001$), but no main effect for genotype ($F_{1,18} = 2.509$, $P = 0.1306$) on change in baseline fEPSPs. The interaction between genotype

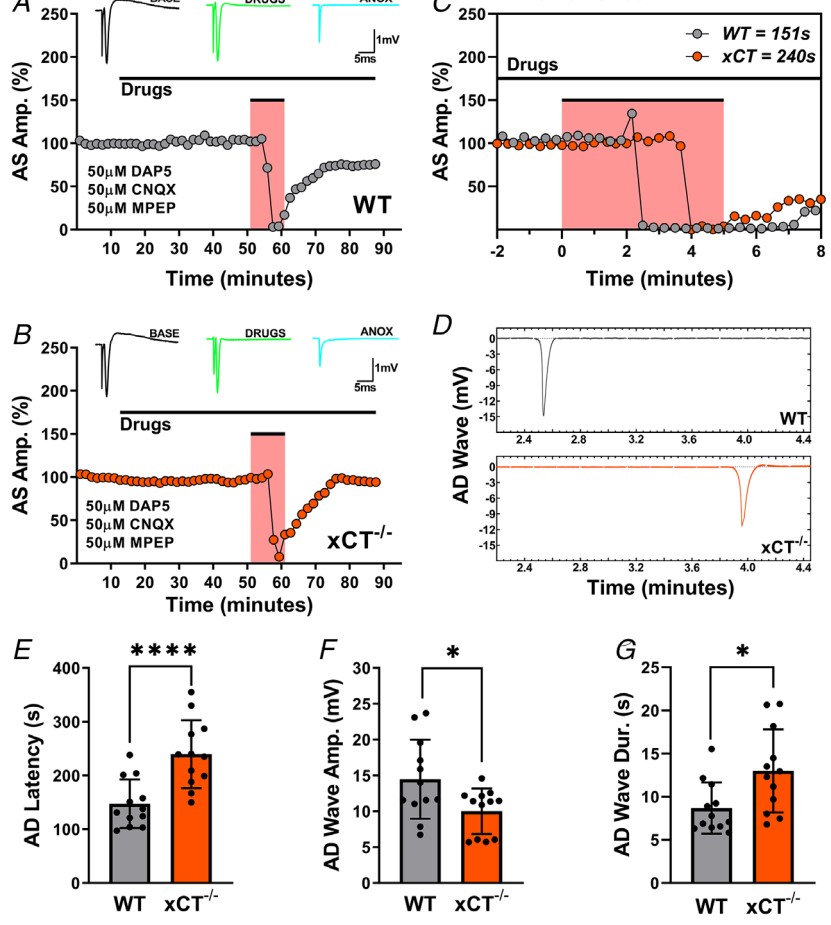

**Figure 5. Glutamate antagonism with low (50 $\mu$M) dosage D-AP5 does not eliminate difference in anoxia tolerance between WT and xCT$^{-/-}$ mice**

*A* and *B*, representative experiment showing antidromic responses for WT (*A*) and xCT$^{-/-}$ (*B*) slices after treatment with 50 $\mu$M CNQX, 50 $\mu$M MPEP and 50 $\mu$M D-AP5. Measurements were recorded from the WT and xCT$^{-/-}$ slice in the same chamber before, during and after an episode of anoxia (pink areas). Each point is an average of five consecutive trials for WT (grey) and xCT$^{-/-}$ (orange) slices. Insets show averaged (*n* = 5) antidromic traces collected during the baseline (BASE), drug perfusion (DRUGS) and anoxic (ANOX) periods. Note that the synaptic component of antidromic spikes were abolished during drug perfusion. *C*, the expanded anoxic period graph illustrating changes in WT and xCT$^{-/-}$ responses with greater temporal resolution. The abscissa shows time relative to anoxia onset. Each point represents a single evoked response for WT (grey) and xCT$^{-/-}$ (orange) synaptic responses. The AD latencies for the WT slice and xCT$^{-/-}$ slice were 151 and 240 s, respectively. *D*, the elimination of the fibre volley and antidromic response was coincident with the appearance of AD waves for both the WT and xCT$^{-/-}$ slice. Time axis relative to anoxia onset as in *C*. *E*–*G*, bar graphs show paired analyses of WT and xCT$^{-/-}$ slices for AD latency, AD wave amplitude and AD wave duration. *P < 0.05, ****P < 0.0001. [Colour figure can be viewed at wileyonlinelibrary.com]

and drug treatment also failed to reach significance ($F_{1,18} = 2.17, P = 0.1573$). The drug had a significant effect on enhancing baseline fEPSPs in WT slices ($P = 0.0017$), but not xCT$^{-/-}$ slices ($P = 0.1541$) (Šidák's test).

Figure 8 summarizes the results from experiments comparing anoxia responses from WT and xCT$^{-/-}$ slices after treatment with 100 $\mu$M CPG. In light of our findings from wash-out experiments (Fig. 4), drug perfusion time

was limited to 2.0 h, and slices were not incubated in the chamber for more than a total of 3.0 h prior to anoxia. In the presence of CPG, the difference in AD latency between WT and xCT$^{-/-}$ slices was eliminated (paired $t_8 = 1.068$, $P = 0.3166$, Fig. 8D), as were the differences in AD wave amplitude (paired $t_8 = 0.937$, $P = 0.375$, Fig. 8E), and AD wave duration (paired $t_8 = 0.6882$, $P = 0.5108$, Fig. 8F). Vehicle-treated slices

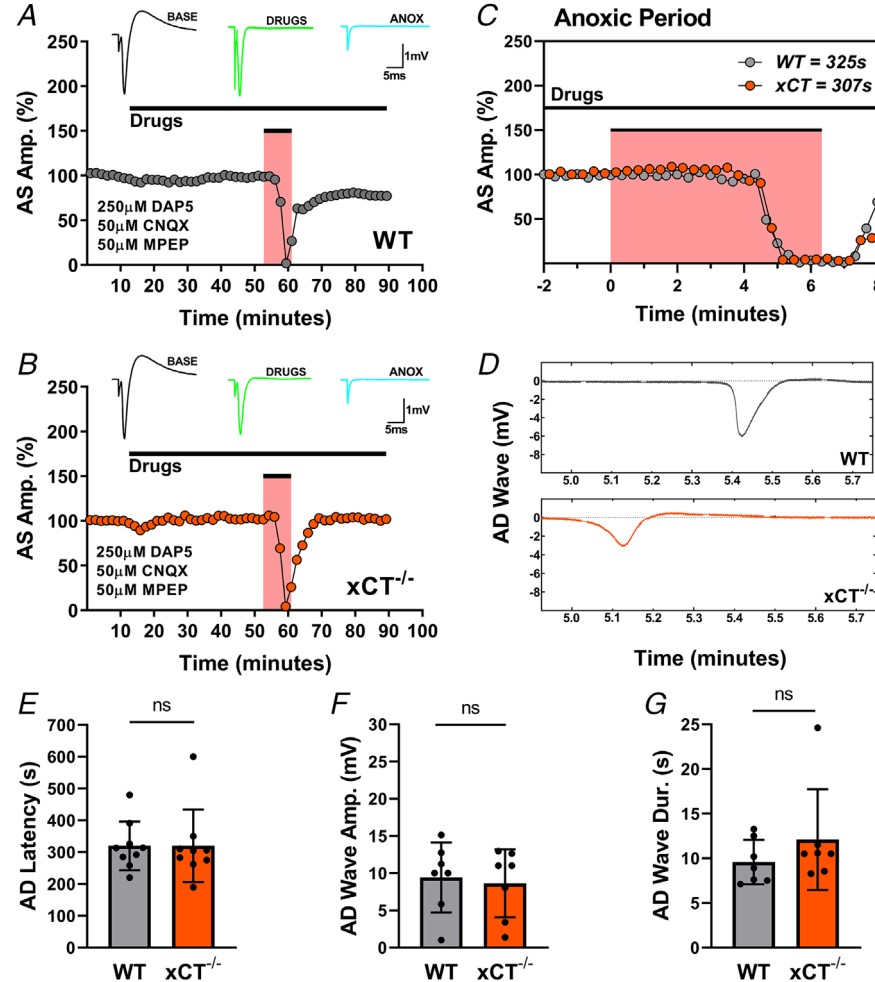

**Figure 6. Glutamate antagonism with high (250 $\mu$M) dosage D-AP5 eliminates difference in anoxia tolerance between WT and xCT$^{-/-}$ slices**

*A* and *B*, representative experiment showing antidromic responses for WT (*A*) and xCT$^{-/-}$ (*B*) slices after treatment with 50 $\mu$M CNQX, 50 $\mu$M MPEP and 250 $\mu$M D-AP5. Measurements were recorded from the WT and xCT$^{-/-}$ slice in the same chamber before, during and after an episode of anoxia (pink areas). Each point is an average of five consecutive trials for WT (grey) and xCT$^{-/-}$ (orange) slices. Insets show averaged ($n = 5$) antidromic traces collected during the baseline (BASE), drug perfusion (DRUGS) and anoxic (ANOX) periods. *C*, the expanded anoxic period graph illustrating changes in WT and xCT$^{-/-}$ responses with greater temporal resolution. The abscissa shows time relative to anoxia onset. Each point represents a single evoked response for WT (grey) and xCT$^{-/-}$ (orange) synaptic responses. The AD latencies for the WT slice and xCT$^{-/-}$ slice were 325 and 307 s, respectively. *D*, the elimination of the fibre volley and antidromic response was coincident with the appearances of AD waves for both the WT and xCT$^{-/-}$ slice. Paired slices showed near simultaneous appearance of AD waves after high D-AP5 dosage. Time axis relative to anoxia onset as in *C*. *E–G*, bar graphs showing paired analyses of WT and xCT$^{-/-}$ slices for AD latency, AD wave amplitude and AD wave duration (AD wave amplitude and duration could not be measured in two experiments). ns, not significant. [Colour figure can be viewed at wileyonlinelibrary.com]

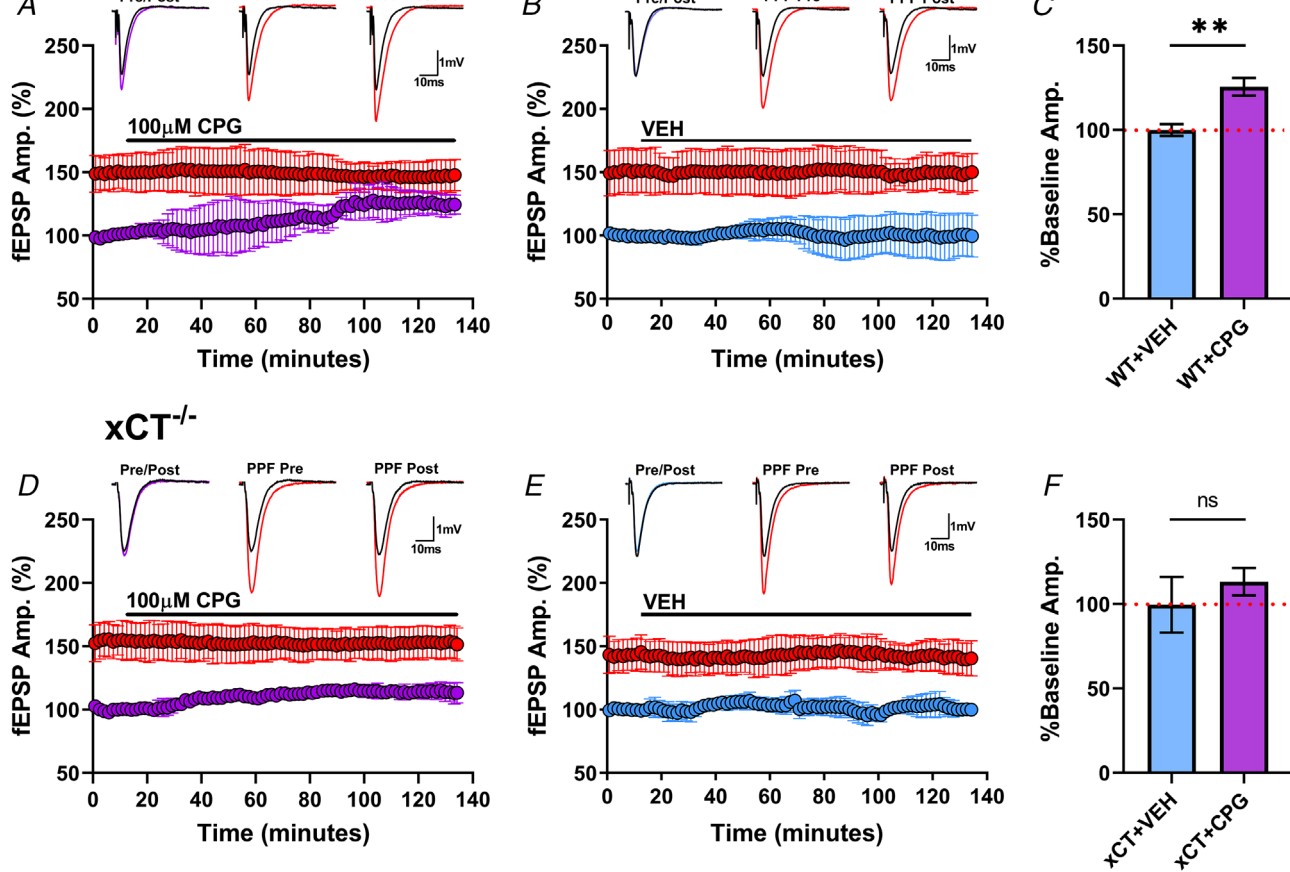

**Figure 7. Effects of CPG administration to WT and xCT$^{-/-}$ slices**

WT and xCT$^{-/-}$ slices were paired in the same chamber at the same time. After a baseline of $\geq$12 min, slices were continuously perfused with 100 $\mu$M CPG or vehicle (VEH) treatment for 2.0 h. *A*, group data from six experiments showing the effect of CPG treatment on fEPSP amplitude and PPF (75 ms IPI) in WT slices. Each point for synaptic (purple circles) and PPF (red circles) responses is an average of five consecutive trials within each experiment, normalized to the baseline average, and averaged (mean $\pm$ SD) across experiments at each time point. PPF values plotted in the graphs are ratios calculated as the amplitude of the second pulse (of the paired pulses) at each time point divided by the amplitude of the first response at the same time point, expressed as a percentage. Insets display representative traces for synaptic response before CPG treatment (black) and after CPG treatment (purple) (Pre/Post) and facilitated (red) responses both before (PPF Pre) and after (PPF Post) CPG treatment. In WT slices, 2.0 h of CPG perfusion elicited a gradual increase of fEPSPs, which stabilized after 90 min at 125.592 $\pm$ 5.227% of baseline responses. Note, the degree of PPF (red) does not change over time even though the amplitude of the first response (purple) does. *B*, group data from five experiments showing the effect of vehicle treatment on fEPSP amplitude and PPF (75 ms IPI) in WT slices. Each point for synaptic (blue circles) and PPF (red circles) responses is an average of five consecutive trials within each experiment, normalized to the baseline average and averaged (mean $\pm$ SD) across experiments at each time point. Insets display representative traces for synaptic response before (black) and after (blue) (Pre/Post) vehicle treatment and facilitated (red) responses both before (PPF Pre) and after (PPF Post) vehicle treatment. Vehicle treatment elicited no appreciable change in baseline fEPSPs for WT slices (99.939 $\pm$ 3.501%). *C*, bar graphs comparing the change in synaptic response for WT slices after 2.0 h of 100 $\mu$M CPG (purple) or vehicle (blue) treatment, measured as the percentage of baseline fEPSP. Red dotted horizontal line highlights 100% of baseline. *D*, group data from six experiments showing the effect of CPG treatment on fEPSP amplitude and PPF (75 ms IPI) in xCT$^{-/-}$ slices. Each point for synaptic (purple circles) and PPF (red circles) responses is an average of five consecutive trials within each experiment, normalized to the baseline average, and averaged (mean $\pm$ SD) across experiments at each time point. Insets display representative traces for synaptic response before CPG treatment (black) and after CPG treatment (purple) (Pre/Post) and facilitated (red) responses both before (PPF Pre) and after (PPF Post) CPG treatment. Perfusion of 100 $\mu$M CPG elicited a gradual increase of fEPSPs, which stabilized after 90 min at 113.244 $\pm$ 8.110% of baseline responses. *E*, group data from five experiments showing the effect of vehicle treatment on fEPSP amplitude and PPF (75 ms IPI) in xCT$^{-/-}$ slices. After a baseline of $\geq$12 min, slices were continuously perfused with vehicle treatment for 120 min. Each point for synaptic (blue circles) and PPF

(red circles) responses is an average of five consecutive trials within each experiment, normalized to the baseline average and averaged (mean ± SD) across experiments at each time point. Insets display representative traces for synaptic response before (black) and after (blue) (Pre/Post) vehicle treatment and facilitated (red) responses both before (PPF Pre) and after (PPF Post) vehicle treatment. Vehicle treatment elicited no appreciable change in baseline fEPSPs for $xCT^{-/-}$ slices ($99.502 ± 16.501\%$). *F*, bar graph comparing the change in synaptic response for $xCT^{-/-}$ slices after 2.0 h of 100 $\mu$M CPG (purple) or vehicle (blue) treatment, measured as the percentage of baseline fEPSP. Red dotted horizontal line highlights 100% of baseline. (paired student's *t* test, ** $P < 0.01$; ns, not significant). [Colour figure can be viewed at wileyonlinelibrary.com]

showed the same WT-$xCT^{-/-}$ differences in AD latency (paired $t_4 = 5.404$, $P = 0.0057$ Fig. 8*K*) and AD wave amplitude (paired $t_4 = 3.654$, $P = 0.0217$, Fig. 8*L*) as in untreated slices, although the difference in AD wave duration did not reach significance (paired $t_4 = 1.735$, $P = 0.1578$, Fig. 8*M*). Anoxic response onset showed no difference between genotypes for either the drug (paired $t_8 = 0.6860$, $P = 0.5121$, Fig. 8*G*) or the vehicle condition (paired $t_4 = 0.000$, $P > 0.999$ Fig. 8*N*).

Figure 9 shows comparisons of the effect of CPG versus vehicle treatment on WT slices (Fig. 9*A–C*) and on $xCT^{-/-}$ slices (Fig. 9*D–F*) from the same dataset in Figure 8. Statistical analyses showed that CPG treatment in WT slices significantly increased AD latency ($t_{12} = 3.305$, $P = 0.0063$, Fig. 9*C*) whereas CPG treatment in $xCT^{-/-}$ slices had no effect ($t_{12} = 0.625$, $P = 0.5389$, Fig. 9*F*). Additionally, a two-way ANOVA was performed with paired slices run as repeated measures for the genotype variable (WT *vs.* $xCT^{-/-}$) and drug treatment (vehicle *vs.* CPG) run as the between-subjects variable. This yielded a significant main effect for genotype ($F_{1,24} = 9.298$, $P = 0.0055$), but no main effect for drug treatment ($F_{1,24} = 3.020$, $P = 0.0950$) on latency to AD. The interaction between genotype and drug treatment was significant ($F_{1,24} = 7.235$, $P = 0.0128$). CPG had a significant lengthening effect on AD latency in WT slices ($P = 0.0269$, Šidák's test), but not $xCT^{-/-}$ slices ($P = 0.9857$). The lack of drug effect in $xCT^{-/-}$ slices strongly supports the conclusion that its action in WT slices was due to inhibition of system $x_c^-$ and not a non-specific drug effect.

## Discussion

The current study undertook a systematic analysis to probe the influence of ambient glutamate during acute stroke onset. Our findings reveal evidence that the cystine/glutamate antiporter, system $x_c^-$, plays a salient role in the ischaemic cascade by regulating tonic levels of extracellular glutamate. Hippocampal slices prepared from mice lacking a functional system $x_c^-$ ($xCT^{-/-}$ mice) showed significantly delayed anoxic depolarization (AD) compared to slices from WT mice. The delayed ADs in $xCT^{-/-}$ mice were reproduced in WT mice by pharmacological inhibition of system $x_c^-$. Manipulations of extracellular glutamate concentrations in slices strongly supported the hypothesis that the delayed ADs in $xCT^{-/-}$ mice were due to the lowered tonic glutamate levels previously observed in these mice *in vivo* (De Bundel et al., 2011) and *in vitro* (Ojeda-Torres et al., 2015). Finally, blockade of glutamate receptors eliminated the differences in AD latency between $xCT^{-/-}$ and WT mice, suggesting that the lower levels of tonic glutamate in $xCT^{-/-}$ mice delay AD by slowing the activation of post-synaptic glutamate receptors during anoxia. These results reinforce, extend, and interpret findings of reduced neuronal death after ischaemic challenge in cell culture systems (Hsieh et al., 2017; Jackman et al., 2012; Soria et al., 2014; Thorn et al., 2015) as well as protection from epileptogenesis (De Bundel et al., 2011; Leclercq et al., 2019; Sears et al., 2019), cerebral ischaemia (Hsieh et al., 2017) and spinal cord injury (Sprimont et al., 2021) *in vivo* when system $x_c^-$ is absent or inhibited.

### Interference with system $x_c^-$ delays anoxic depolarization

Our experiments employed a well-characterized hippocampal slice model in which oxygen deprivation (anoxia) is used to simulate stroke (Arrigoni et al., 2005; Coelho et al., 2006; Croning & Haddad, 1998; Dale et al., 2000; Fischer et al., 2009; Heit et al., 2021; Lipton, 1999; Pearson & Frenguelli, 2004; Wang et al., 1999). Prior work has established that anoxia results in a neuronal depolarization mediated by rundown of membrane pumps, acceleration of action potential-dependent synaptic glutamate release, and further depolarization due to activation of glutamate receptors. These events culminate in anoxic depolarization (AD), signalled by an extracellular slow wave (AD wave) and loss of neuronal membrane potentials. Our experiments show a significant delay in the appearance of AD in mice lacking system $x_c^-$; furthermore, the extracellular currents underlying AD (the AD waves) were flattened, being reduced in amplitude and longer in duration, compared to WT mice. Moreover, PPF magnitude was equivalent between genotypes, and therefore the ischaemic neuroprotection observed in $xCT^{-/-}$ mice cannot be explained by decreased pre-synaptic calcium loading and/or decreased glutamate release probability. Similar differences in AD latency and AD wave shape were observed in comparisons of $xCT^{-/-}$ and WT slices from mice at 2–4 months and

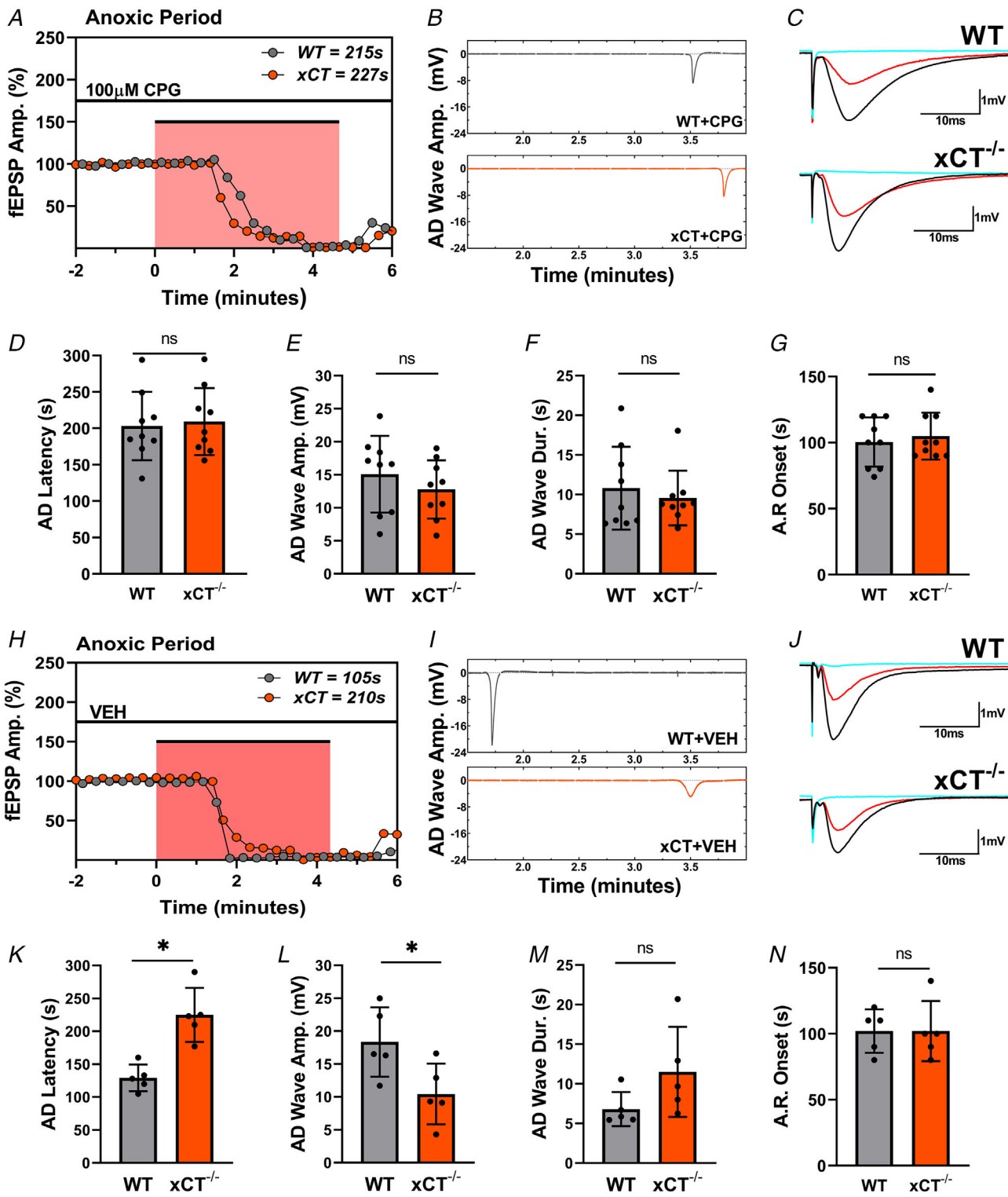

**Figure 8. Pharmacological antagonism of system $x_c^-$ in WT slices reproduces effect of xCT KO on anoxia tolerance**

*A*, representative anoxic period graph highlighting changes in WT (grey) and xCT$^{-/-}$ (orange) synaptic responses after 2-h treatment with CPG (100 $\mu$M), where each point represents a single evoked response. The abscissa shows time relative to anoxia. For this paired experiment, the WT slice recorded an AD latency of 215 s, and the xCT$^{-/-}$ slice recorded an AD latency of 227 s. *B*, the elimination of the fibre volley and synaptic response was coincident with the appearances of AD waves for both the WT and xCT$^{-/-}$ slice. Time axis relative to anoxia

onset as in *A*. Paired slices showed near simultaneous appearance of AD waves after CPG treatment. *C*, waveform traces show averaged (*n* = 5) individual responses during baseline (black), anoxic (blue) and recovery (red) periods for WT and xCT$^{-/-}$ slice. *D–G*, bar graphs showing paired analyses of WT and xCT$^{-/-}$ slices for AD latency, AD wave amplitude, AD wave duration, and anoxic response onset. *H*, representative anoxic period graph highlighting changes in WT (grey) and xCT$^{-/-}$ (orange) synaptic responses after treatment with CPG vehicle (VEH), where each point represents a single evoked response. The abscissa shows time relative to anoxia. For this paired experiment, the WT slice recorded an AD latency of 105 s, and the xCT$^{-/-}$ slice recorded an AD latency of 210 s. *I*, the elimination of the fibre volley and synaptic response was coincident with the appearances of AD waves for both the WT and xCT$^{-/-}$ slice. Time axis relative to anoxia onset as in *H*. *J*, waveform traces show averaged (*n* = 5) individual responses during baseline (black), anoxic (blue) and recovery (red) periods for WT and xCT$^{-/-}$ slice. *K–N*, bar graphs showing paired analyses of WT and xCT$^{-/-}$ slices for AD latency, AD wave amplitude, AD wave duration and anoxic response onset. (paired student's *t* test, \*$P < 0.05$; ns, not significant). [Colour figure can be viewed at wileyonlinelibrary.com]

12–18 months of age, indicating that the influence of system x$_c^-$ on AD is robust and stable over much of the lifespan. Comparatively, system x$_c^-$ deletion in aged mice has also been shown to generate health- and lifespan extension, provide prophylaxis against age-related decline of hippocampal-dependent memory, and shift the aged hippocampus toward a more anti-inflammatory state (Verbruggen et al., 2022).

To further test the role of system x$_c^-$ in AD, we used an inhibitor of the antiporter. Two hours of incubation with CPG significantly increased AD latencies in WT, but not xCT$^{-/-}$ mice. Paired experiments testing WT and xCT$^{-/-}$ slices subjected to anoxia in the same chamber showed no differences in AD latency in the presence of CPG. Control experiments using the drug vehicle reproduced the increased latency in xCT$^{-/-}$ slices over WT slices seen in untreated preparations. In addition to inhibition of system x$_c^-$, CPG has a number of other actions, including antagonism of group I metabotropic glutamate receptors (mGluRs). However, the lack of drug effect in xCT$^{-/-}$ slices strongly suggests that CPG's influence on AD latency is due to its action on system x$_c^-$, which ostensibly decreased tonic glutamate levels prior to anoxia, thus mitigating excitotoxicity.

The first electrophysiological sign of anoxia in hippocampal slices is a suppression of synaptic transmission. This suppression results from the release of adenosine into the extracellular space as a consequence of ATP breakdown; adenosine inhibits synaptic transmission by its action on presynaptic A$_1$ receptors (Dunwiddie & Masino, 2001; Fowler, 1989; Heit et al., 2021). This anoxic response onset was not detectably different in xCT$^{-/-}$ *versus* WT slices, suggesting that the initial metabolic effects of anoxia were not affected by loss of system x$_c^-$. Similar results were obtained with CPG: the drug had no effect on anoxic response onset in either xCT$^{-/-}$ or WT slices.

A previous study (Soria et al., 2014) reported a somewhat different pattern of responses after CPG treatment during combined oxygen and glucose deprivation; the drug decreased AD wave amplitudes, but did not affect AD latency. This investigation, however,

recorded voltage-clamp currents from cultured slices in cortical layer V of P18–P30 Sprague–Dawley rats for these measurements, and only provided 1 h of CPG pre-treatment – a potentially insufficient epoch for system x$_c^-$ antagonism to thoroughly decrease ambient glutamate levels in the extracellular space. Further work will be needed to determine whether these differences are due to developmental factors, regional differences or the methods used to induce AD.

Importantly, pre-existing lines of evidence suggest tonic glutamate can act in paracrine fashion to suppress glutamatergic synapse strength by triggering removal of postsynaptic glutamate receptors. In our experiments, 2 h of system x$_c^-$ inhibition via CPG treatment produced a significant enhancement of fEPSPs in WT, but not xCT$^{-/-}$, slices. These data support the notion that xCT-mediated alterations of CA1 synapses involve mGluRs coupled to pathways, which modulate post-synaptic AMPAR abundance (Williams & Feathersrone, 2014). Specifically, ambient glutamate can activate extrasynaptic Group 1 mGluRs, which subsequently elicit a cascade of intracellular events resulting in reduced postsynaptic glutamate receptor expression and altered trafficking (Gladding et al., 2009; Oliet et al., 1997; Palmer et al., 1997). In our CPG experiments, ambient glutamate levels were reduced in WT slices as a consequence of system x$_c^-$ antagonism, thereby inhibiting the activation of mGluRs and hence their removal of post-synaptic AMPARs. CPG treatment therefore resulted in increased AMPAR-mediated synaptic transmission, and yet paradoxically increased AD latencies, further highlighting the excitatory salience of tonic glutamate during oxygen-depletion events.

### The role of ambient extracellular glutamate in AD

System x$_c^-$ resides in astrocytes, where the antiporter imports cystine and exports glutamate in exchange (Conrad & Sato, 2012), serving as a non-synaptic, calcium-independent source of glutamate. Microdialysis studies indicate that the action of system x$_c^-$ has a significant impact on extracellular concentrations of

glutamate: in hippocampal slices, extracellular glutamate is 60% higher in WT *versus* xCT$^{-/-}$ mice (Ojeda-Torres et al., 2015). When we clamped the levels of extracellular glutamate by perfusing slices with ACSF containing 5 $\mu$M glutamate, the concentration measured in WT slices, anoxia resulted in xCT$^{-/-}$ slices reaching AD at the shorter latency observed in WT slices. This manipulation also equalized the AD wave amplitude and duration in WT and xCT$^{-/-}$ mice. In a second

experiment, we allowed the elevated glutamate to wash out from WT slices by extended (6–7 h) perfusion with glutamate-free ACSF (Ojeda-Torres et al., 2015; Williams & Featherstone, 2014). This manipulation caused an increase in AD latency in WT slices so that they were equivalent to AD latencies in xCT$^{-/-}$ slices. AD wave amplitude and duration were also normalized between genotypes by glutamate wash-out. Taken together, these findings support the hypothesis that system x$_c^-$ regulates

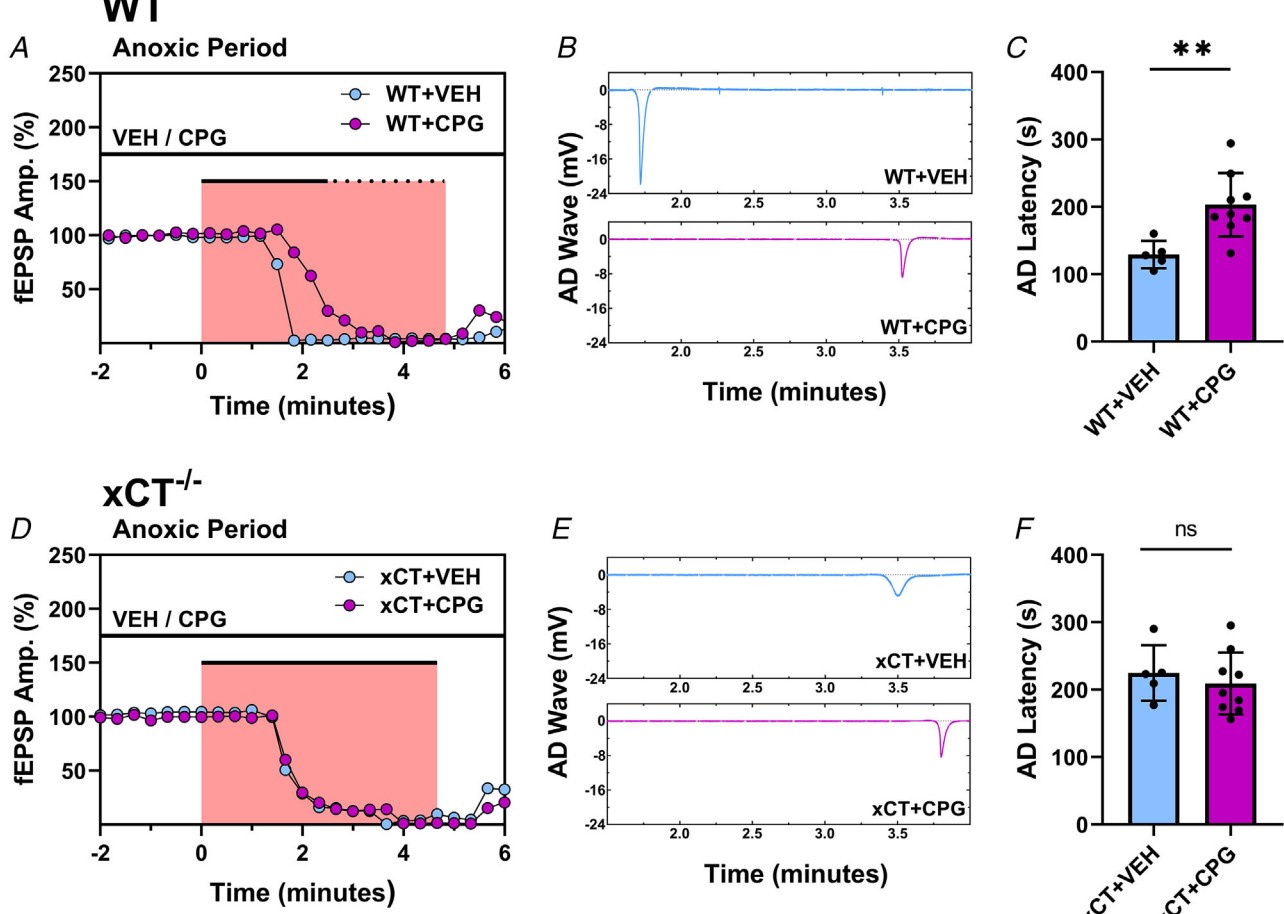

**Figure 9. Comparisons of effects of CPG *versus* vehicle on WT and xCT$^{-/-}$ slices**
*A*, graph showing individual responses from WT slices during the anoxia period for vehicle (WT+VEH) and CPG (WT+CPG) conditions. The abscissa shows time relative to anoxia onset normalized between experiments. Each point represents a single evoked synaptic response from the vehicle- (blue) and CPG-treated (purple) WT slice, and the pink area denotes the anoxic period. The bars above the pink area delineate the total anoxic period (before re-oxygenation) for the vehicle (continuous bar) and CPG (continuous plus dotted bar) condition. The AD latency for the vehicle-treated WT slice was 105 s and the AD latency for the CPG-treated WT slice was 215 s. *B*, panels showing AD waves during the vehicle (WT+VEH) and CPG (WT+CPG) conditions. The time axis is relative to anoxia onset as in *A*. *C*, bar graph comparing group data of WT AD latencies after vehicle and CPG administration. *D*, graph showing individual responses from xCT$^{-/-}$ slices during the anoxia period for vehicle (xCT+VEH) and CPG (xCT+CPG) conditions. The abscissa shows time relative to anoxia onset normalized between experiments. Each point represents a single evoked synaptic response from the vehicle- (blue) and CPG-treated (purple) xCT$^{-/-}$ slice, and the pink area denotes the anoxic period. The AD latency for the vehicle-treated xCT$^{-/-}$ slice was 210 s and the AD latency for the CPG-treated xCT$^{-/-}$ slice was 227 s. *E*, panels showing AD waves during the vehicle (xCT+VEH) and CPG (xCT+CPG) conditions. The time axis is relative to anoxia onset as in *D*. *F*, bar graph comparing group data of xCT$^{-/-}$ AD latencies after vehicle and CPG administration. (paired student's *t* test, **$P < 0.01$; ns, not significant).
[Colour figure can be viewed at wileyonlinelibrary.com]

the timing and magnitude of AD generation via its regulation of ambient extracellular glutamate levels. In like manner, tonic extracellular glutamate has also been recently implicated in the spreading depolarizations accompanying migraine aura (Parker et al., 2021).

In addition to system $x_c^-$, extracellular glutamate levels in the hippocampus are regulated by astrocytic glutamate transporters, primarily EAAT1 (GLAST) and EAAT2 (GLT-1), which are believed to rapidly remove synaptic glutamate released during synaptic transmission (Danbolt, 2001). There is no evidence for altered expression of these transporters in $xCT^{-/-}$ mice (De Bundel et al., 2011), indicating that the altered ambient glutamate levels measured in these mice are due to absence of a functional system $x_c^-$. The absolute concentration of extracellular glutamate in the brain is a matter of debate. Estimates based on tonic activation of synaptic NMDARs (Herman & Jahr, 2007; Le Meur et al., 2007) are much lower than those measured using microdialysis or push–pull perfusion (Montiel et al., 2005; Ojeda-Torres et al., 2015). The contribution of system $x_c^-$ to the former measures is unknown. Nevertheless, our results show that experimental manipulation of extracellular glutamate concentrations differentially affects AD latency in $xCT^{-/-}$ and WT slices, most parsimoniously explained by the antiporter's effects on ambient glutamate levels.

### Tonic glutamate, NMDA receptors and AD

Previous studies have extensively described the importance of glutamate excitotoxicity in cell death after ischaemia (Choi, 1992; Lipton, 1999), but few have probed the effect of glutamate receptors in the timing of AD (Heit et al., 2021). Hippocampal neurons in $xCT^{-/-}$ mice display enhanced AMPAR-mediated synaptic currents and increased expression of AMPAR proteins, with no evidence of changes in GABAergic synaptic transmission (Williams & Featherstone, 2014). Our experiments with glutamate receptor antagonists showed that NMDARs rather than AMPARs critically regulate the differences in timing of AD in WT and $xCT^{-/-}$ slices. In the first experiment, we used concentrations of CNQX (50 $\mu$M) and D-AP5 (50 $\mu$M) sufficient to abolish synaptic transmission mediated by AMPA and NMDA receptors, respectively, under normoxic conditions. These antagonists had only marginal effects on AD latency and did not eliminate the differences between $xCT^{-/-}$ and WT slices in AD latency or AD wave amplitude and duration. We reasoned that the massive release of glutamate during anoxia might outcompete D-AP5 at NMDARs; therefore, in the second experiment, we increased the concentration of D-AP5 five-fold, while keeping the CNQX concentration the same as before. Under these conditions, WT AD latencies were significantly increased,

and the differences between $xCT^{-/-}$ and WT slices in AD latency, AD wave amplitude and AD wave duration were eliminated. These results support prior findings showing that the activation of NMDARs is critical for the timing of AD in hippocampus (Fusco et al., 2018; Heit et al., 2021; Tanaka et al., 1997). The equivalence of AD latency in $xCT^{-/-}$ and WT slices with glutamate receptors blocked suggests that the influence of system $x_c^-$ on the timing of AD is mediated by NMDARs. Alternatively, system $x_c^-$ might have an action on AD latency that is independent of NMDARs but is obscured when those receptors are blocked. In this case, the equivalence of AD latency in $xCT^{-/-}$ and WT slices under NMDAR blockade would reflect a kind of ceiling effect.

If we accept the simpler interpretation, that system $x_c^-$ affects AD latency via an effect mediated by NMDARs, the likely mechanism would involve the antiporter's influence on tonic glutamate levels. AD would be faster in WT mice since the higher levels of tonic glutamate would magnify or synergize with the anoxia-dependent glutamate release to accelerate AD. Mice lacking system $x_c^-$ ($xCT^{-/-}$ mice) would have delayed AD due to their lower levels of tonic glutamate. Two alternative mechanisms are possible, but less likely: first, system $x_c^-$ is upregulated during anoxic conditions (Hsieh et al., 2017) and elevated glutamate release in exchange for cystine could contribute to AD; however, the upregulation does not appear to be rapid enough to affect AD latency (Thorn et al., 2015). Second, the higher tonic glutamate levels in mice with functional system $x_c^-$ could modulate the number or sensitivity of NMDARs such that they respond more quickly to glutamate released during anoxia; however, there is no direct evidence for an alteration in NMDARs in $xCT^{-/-}$ mice. As noted above, synapses in $xCT^{-/-}$ mice have a higher abundance of AMPARs that produce larger postsynaptic currents in response to synaptic glutamate release (Williams & Featherstone, 2014); more AMPARs would tend to accelerate AD rather than delay it. That being said, a specific change in extrasynaptic NMDARs as a consequence of system $x_c^-$ activity remains plausible (Soria et al., 2014). In fact, the tonic activation of NMDARs by ambient glutamate (Dalby & Mody, 2003; Sah et al., 1989) can enhance cell excitability by increasing the proclivity toward regenerative depolarization (Cavelier et al., 2005). It is therefore possible that a population of NMDARs are constitutively activated by xCT-mediated glutamate levels, and that this tonic conductance sensitizes neurons to ischaemia-induced glutamate excitotoxicity.

To the authors' knowledge, this is the first report suggesting genetic deletion or pharmacological inhibition of system $x_c^-$ increases AD latency due to the antiporter's contribution to ambient extracellular glutamate levels. xCT-mediated glutamate release enhances both the speed and synchrony of depolarizing events during anoxia, and

thus may exacerbate the generation of the ischaemic core, which is clinically defined (Nakamura et al., 2010) and diagnosed (Ayata, 2018; Hartings et al., 2017) as the region of anoxic depolarization. As such, inhibition of the antiporter could be considered as a novel target for stroke therapeutics in lieu of synaptic glutamate antagonism. Perhaps the central upshot of the current data is to focus our efforts on antagonizing the source(s) of glutamate as opposed to the site of its action. Although system $x_c^-$ inhibition may not prevent AD in the ischaemic core, the delayed occurrence and desynchronization of AD waves may ameliorate the development of intractable pathology in areas downstream of the ischaemic focus, which constitute the ischaemic penumbra. All things considered, the effect of system $x_c^-$ on both physiological and pathological glutamatergic neurotransmission is an area ripe for exploration; and the findings herein elucidate the antiporter's involvement in the noxious, glutamate-driven events of the ischaemic cascade.

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

## Additional information

### Data availability statement

All data reported in this investigation are represented in the manuscript figures and/or are available upon request. Code for the stimulation protocols employed are available upon reasonable request.

### Competing interests

None.

### Author contributions

J.L.: conception or design of the work; acquisition, analysis or interpretation of data for the work; drafting the work or revising it critically for important intellectual content; final approval of the version to be published; agreement to be accountable for all aspects of the work. B.H.: conception or design of the work; drafting the work or revising it critically for important intellectual content; final approval of the version to be published; agreement to be accountable for all aspects of the work. A.C.: acquisition, analysis or interpretation of data for the work; drafting the work or revising it critically for important intellectual content; final approval of the version to be published; agreement to be accountable for all aspects of the work. A.S.: acquisition, analysis or interpretation of data for the work; drafting the work or revising it critically for important intellectual content; final approval of the version to be published; agreement to be accountable for all aspects of the work. D.F.: conception or design of the work; drafting the work or revising it critically for important intellectual content; final approval of the version to be published; agreement to be accountable for all aspects of the work. T.P.: conception or design of the work; drafting the work or revising it critically for important intellectual content. All authors have approved the final version of the manuscript and agree to be accountable for all aspects of the work. All persons designated as authors qualify for authorship, and all those who qualify for authorship are listed.

### Funding

This work was supported by the National Science Foundation (IOS grant no. 1655494).

### Dedication

This work is dedicated to Dr David E. Featherstone of the University of Illinois at Chicago (deceased 28 January 2017). Dave was a rigorous and innovative scientist, as well as an adroit and beloved professor, who never displayed an ounce of hubris. His influence as a researcher and educator will reverberate to generations of neuroscientists.

### Author's present address

B. S. Heit: Department of Neuroscience and Department of Biomedical Engineering, Northwestern University, Chicago, IL 60611, USA

### Keywords

ambient glutamate, cystine/glutamate transporter, excitotoxicity, stroke

## Supporting information

Additional supporting information can be found online in the Supporting Information section at the end of the HTML view of the article. Supporting information files available:

**Statistical Summary Document**
**Peer Review History**

