## [Peer Review History · The Journal of Physiology]

Tonic extracellular glutamate and ischemia: glutamate antiporter system xc⁻ regulates anoxic depolarization in hippocampus

John R Larson, Bradley Stavros Heit, Alex Chu, Abhay Sane, David E Featherstone, and Thomas J Park
DOI: 10.1113/JP283880

Corresponding author(s): John Larson (jrl@uic.edu)

The following individual(s) involved in review of this submission have agreed to reveal their identity: Patrice Quintana (Referee #1)

Review Timeline:

Submission Date:	26-May-2022
Editorial Decision:	21-Jun-2022
Resubmission Received:	27-Sep-2022
Accepted:	18-Oct-2022

Senior Editor: Katalin Toth

Reviewing Editor: Samuel Young

Transaction Report:

Dear Dr Larson,

Re: JP-RP-2022-283336 "Tonic extracellular glutamate and ischemia: glutamate antiporter system xc- regulates anoxic depolarization in hippocampus" by John R Larson, Bradley Stavros Heit, Alex Chu, Abhay Sane, David E Featherstone, and Thomas J Park

Thank you for submitting your manuscript to The Journal of Physiology. It has been assessed by a Reviewing Editor and by 2 Referees and the reports are copied below.

Please let your co-authors know of the following editorial decision as quickly as possible.

As you will see, in its current form, the manuscript is not acceptable for publication in The Journal of Physiology. In comments to me, the Reviewing Editor expressed interest in the potential of this study, but much work still needs to be done (and this may include new experiments) in order to satisfactorily address the concerns raised in the reports.

In view of this interest, I would like to offer you the opportunity to carry out all of the changes requested in full, and to resubmit a new manuscript using the "Submit Special Case Resubmission for JP-RP-2022-283336..." on your homepage.

We cannot, of course, guarantee ultimate acceptance at this stage as the revisions required are substantial. However, we encourage you to consider the requested changes and resubmit your work to us if you are able to complete or address all changes.

A new manuscript would be renumbered and redated, but the original referees would be consulted wherever possible. An additional referee's opinion could be sought, if the Reviewing Editor felt it necessary. A full response to each of the reports should be uploaded with a new version.

I hope that the points raised in the reports will be helpful to you.

Yours sincerely,

Katalin Toth
Senior Editor
The Journal of Physiology

EDITOR COMMENTS

Reviewing Editor:

This manuscript provides novel findings concerning the role of the Xc-cysteine/glutamate antiporter in regulating anoxic depolarization. The experiments were designed and carefully executed. Despite the enthusiasm for the manuscript both reviewers had issues in the manuscript that need to be addressed. Both reviewers had concerns about Figure 1 in regards to fiber volley, as changes in fibers volley is a key sign of anoxic depolarization. Another concern was about the information regarding the health and quality control of acute slices used for experimental data. This lack of information on quality control is critical as the health of the tissue used will have significant impact on the conclusions that can be drawn. Reviewer#1 had concern about potential off target effects of CPG. Reviewer#2 also pointed out that in Figure 1G data is inconsistent with Figure 7K data and this needs to be addressed. All in all, these points need to be addressed. Finally, the authors should carefully revise and rewrite the manuscript in response to both the reviewer's constructive comments.

REFEREE COMMENTS

Referee #1:

In the paper « Tonic extracellular glutamate and ischemia : glutamate antiporter system xc- regulates anoxic depolarization in hippocampus » the authors investigated the involvement of cystine/glutamate antiporter xc- in anoxic depolarization following an anoxic episode. Focusing on the possible involvement of glutamate present tonically in extracellular space in deleterious postischemic mechanisms is an interesting approach.

To this end they used field recording electrophysiologic technique, mutant mice lacking xc-, its specific antagonist and pharmacologic tools.

The authors show that, when subjected to a transient anoxic period, fEPSP recorded in hippocampal slices from both young and aged xCT-/- mice display a longer AD latency, attenuated AD wave amplitude but increased duration compared to WT. Recording in parallel WT and xCT-/- slices exposed to anoxia within the same chamber is a clever approach.

By pharmacologically blocking xCT, they also reinforce the idea of an implication of this antiport by obtaining results similar to those found with xCT-/- mice.

With the experiments on restoration of ambient glutamate in xCT-/- slices or washout of ambient glutamate in WT slices they demonstrate the involvement of tonic extracellular glutamate in AD latency.

1) In normoxic conditions there is apparently no alteration of the probability of vesicular release at the synapse in xCT-/- mice as demonstrated by PPF. It would have been interesting to perform such a test during and after the anoxic episode to be sure that xCT-/- mutation has also no influence on this short term plasticity mechanism in pathologic conditions.

2) In the inserts Fig 1 A, B and I where the illustrations are averages of 5 individual responses, what do we see just before the EPSP : the stimulation artifact or the fiber volley ? Same remark for illustrations in Fig 2, 4, 5 and 6. It would have been interesting to see illustrations of responses in Fig 3. After 6-7 hours of incubation followed by anoxia and measurement of AD latency, I am curious to know what state the acute hippocampal slices were in. It must have been hard to get valid fEPSPs. Probably not the easiest part of the investigation. It would be interesting to check the evolution of the post-synaptic response to pre-synaptic fibre volley ratio before and after the anoxic episode to see if, in this model, some deleterious effects of the anoxic episode on the integrity of the neuronal network occur within the following post-anoxic hour.

3) Effect of CPG administration (Fig6) : if I have correctly understood, the authors performed PPF (75ms IPI) in WT and xCT-/- slices before and during CPG administration, or sham. In Fig6 A, B, C and D white circles are averages of the first EPSP of the PPF, red circles of the second one (50% greater). Both are normalized to the initial baseline average.

First, the colors of traces pre/post are the same as for the illustration of the traces of the PPF, it's quite confusing, blue and purple would be more coherent with Illustrations C and F. Secondly, if there is an increase of the amplitude of the first EPSP (by 25% in WT and 13% in xCT-/- mice) but not of the second, as illustrated in Fig6 A, C, D and F, PPF ratio might be modified. In Fig6 A traces in the inserts don't give such impression, red ones (second EPSP of the PPF, normalized to initial baseline average) are greater after CPG administration. Please give an explanation. A timelapse evolution of PPF ratio before and during CPG perfusion would help to see more clearly its evolution. If there is a modification please give an explanation. Thirdly, the authors use CPG (S-4-carboxphenylglycine) as system xC- inhibitor. However, CPG is known as a competitive group I metabotropic glutamate receptor antagonist. Is there another, more specific, inhibitor of system xC- to perform the same tests? If not additional experiments with another mGluR I antagonist are necessary, AIDA for instance, to verify or clarify the degree of possible involvement of metabotropic receptors.

4) As clearly illustrated in Fig1A, B and Fig2A, B, D, E, there is systematically (?) a lasting depression of fEPSP amplitudes after the anoxic episode in both WT and xCT-/- mice. But this lasting depression seems to be less important in xCT-/- mice. If so, additional experiments using others technical approaches (WB of glutamate receptors subunits, their phosphorylated / unphosphorylated forms, before and after anoxia for instance) would be necessary to investigate system xC- involvement on such lasting modifications.

5) Pages 4 and 27: the authors present the model of hippocampal slices subjected to oxygen deprivation as an extensively used in vitro model of stroke. To support this assertion they cite three references. The Croning and Haddad 1998 reference is just a comparison of brain slice chamber designs (submerged and interface) for investigations of oxygen deprivation in vitro, interestingly glucose deprivation is also used in some of the experiments, pointing the interest about glucose levels. The second citation is the excellent and complete review of P. Lipton on ischemic cell death in neurons, published in 1999. It's interesting to note that P. Lipton has widely used the oxygen/glucose deprivation as an in vitro model of stroke in his papers. The authors also cite Heit et al. 2021 work, which is the only of the three references where anoxia alone is used...

Referee #2:

In this study, Heit et al investigated the role of system Xc- cystine/glutamate antiporter in the progression of anoxic depolarization (AD) caused by oxygen deprivation during ischemic stroke. Stroke is a leading cause of death and disability worldwide with limited treatment. AD is induced by the loss of membrane potential in neurons due to a cascade of cellular events including an increase of glutamate release, leading to excitotoxicity and neuronal death. System Xc- transporter mediates the exchange of extracellular cystine and intracellular glutamate across the cellular plasma membrane and thereby contributes to most of the tonic glutamate release. With electrophysiological recordings from hippocampal slices in an in vitro anoxia model, the authors showed that genetic deletion (with xCT knockout mice) or pharmacological inhibition (with a compound CPG) of system Xc- increased AD latency and altered AD waves. By manipulating the extracellular glutamate concentration or blocking glutamate receptors, they further demonstrated that the protective effect of system Xc- deletion was likely attributed to a reduced level of tonic glutamate that could activate NMDA receptors and accelerate AD. Overall, the experiments were well designed and carefully executed. The findings are novel and have clinical implications. However, the manuscript can be improved if certain results are better interpreted and discussed.

Major comments:

1. In Figure 1 A and 1B, the amplitude of fEPSP at the baseline appeared comparable between the WT and xCT^{-/-} groups. However, the fiber volley preceding the xCT^{-/-} fEPSP was much bigger. Is there a group difference in the amplitude of fiber volley? If yes, this needs to be clarified because the change in the fiber volley is one of the critical signs for AD, as the authors pointed out. If there is no group difference, the authors may choose better representative traces.
2. In Figure 7K, the AD wave duration was not different between the WT and xCT^{-/-} groups under the Sham condition. This is contradictory to the result shown in Figure 1G. Was this because the sample size for Figure 7K was too small? If yes, the authors may consider increasing the sample size.
3. Only male mice were used in this study. Please justify the choice of sexes.
4. A prolonged exposure (6-7 hours) to ACSF might deteriorate the overall health of brain slices. It would be helpful to briefly describe quality control used for recordings in the Materials and Methods section.
5. The system Xc- antagonist (CPG) increased AMPA receptor-mediated synaptic potential in WT neurons, but not in xCT^{-/-} neurons (Figure 6). This difference requires more explanation.

Minor comments:

1. In all relevant figures, please change "Sham" to "Vehicle" to be consistent with the text.
2. Page 4, please define the abbreviation of "mEPSC" and "sEPSC".
3. Page 5, remove "of" from "...Committee at of the University...".
4. Page 6, please define "DC" in "...a large slow DC shift...".
5. Page 7, add "the" in the sentence "...without added drug"?

ADDITIONAL FORMATTING REQUIREMENTS:

-Author photo and profile. First (or joint first) authors are asked to provide a short biography (no more than 100 words for one author or 150 words in total for joint first authors) and a portrait photograph. These should be uploaded and clearly

labelled with the revised version of the manuscript. See Information for Authors for further details.

-You must start the Methods section with a paragraph headed Ethical Approval. A detailed explanation of journal policy and regulations on animal experimentation is given in Principles and standards for reporting animal experiments in The Journal of Physiology and Experimental Physiology by David Grundy J Physiol, 593: 2547-2549. doi:10.1113/JP270818.). A checklist outlining these requirements and detailing the information that must be provided in the paper can be found at: <https://physoc.onlinelibrary.wiley.com/hub/animal-experiments>. Authors should confirm in their Methods section that their experiments were carried out according to the guidelines laid down by their institution's animal welfare committee, and conform to the principles and regulations as described in the Editorial by Grundy (2015). The Methods section must contain details of the anaesthetic regime: anaesthetic used, dose and route of administration and method of killing the experimental animals.

-Your manuscript must include a complete Additional Information section

-Please upload separate high-quality figure files via the submission form.

-Please ensure that the Article File you upload is a Word file.

-A Statistical Summary Document, summarising the statistics presented in the manuscript, is required upon revision. It must be on the Journal's template, which can be downloaded from the link in the Statistical Summary Document section here: https://jp.msubmit.net/cgi-bin/main.plex?form_type=display_requirements#statistics

-Papers must comply with the Statistics Policy https://jp.msubmit.net/cgi-bin/main.plex?form_type=display_requirements#statistics

In summary:

-If $n \leq 30$, all data points must be plotted in the figure in a way that reveals their range and distribution. A bar graph with data points overlaid, a box and whisker plot or a violin plot (preferably with data points included) are acceptable formats.

-If $n > 30$, then the entire raw dataset must be made available either as supporting information, or hosted on a not-for-profit repository e.g. FigShare, with access details provided in the manuscript.

- n clearly defined (e.g. x cells from y slices in z animals) in the Methods. Authors should be mindful of pseudoreplication.

-All relevant n values must be clearly stated in the main text, figures and tables, and the Statistical Summary Document (required upon revision)

-The most appropriate summary statistic (e.g. mean or median and standard deviation) must be used. Standard Error of the Mean (SEM) alone is not permitted.

-Exact p values must be stated. Authors must not use 'greater than' or 'less than'. Exact p values must be stated to three significant figures even when 'no statistical significance' is claimed.

-Statistics Summary Document completed appropriately upon revision

-A Data Availability Statement is required for all papers reporting original data. This must be in the Additional Information section of the manuscript itself. It must have the paragraph heading "Data Availability Statement". All data supporting the results in the paper must be either: in the paper itself; uploaded as Supporting Information for Online Publication; or archived in an appropriate public repository. The statement needs to describe the availability or the absence of shared data. Authors must include in their Statement: a link to the repository they have used, or a statement that it is available as Supporting Information; reference the data in the appropriate section(s) of their manuscript; and cite the data they have shared in the References section. Whenever possible the scripts and other artefacts used to generate the analyses presented in the

paper should also be publicly archived. If sharing data compromises ethical standards or legal requirements then authors are not expected to share it, but must note this in their Statement. For more information, see our Statistics Policy.

-Please include an Abstract Figure. The Abstract Figure is a piece of artwork designed to give readers an immediate understanding of the research and should summarise the main conclusions. If possible, the image should be easily 'readable' from left to right or top to bottom. It should show the physiological relevance of the manuscript so readers can assess the importance and content of its findings. Abstract Figures should not merely recapitulate other figures in the manuscript. Please try to keep the diagram as simple as possible and without superfluous information that may distract from the main conclusion(s). Abstract Figures must be provided by authors no later than the revised manuscript stage and should be uploaded as a separate file during online submission labelled as File Type 'Abstract Figure'. Please ensure that you include the figure legend in the main article file. All Abstract Figures should be created using BioRender. Authors should use The Journal's premium BioRender account to export high-resolution images. Details on how to use and access the premium account are included as part of this email.

Confidential Review

26-May-2022

In this study, Heit et al investigated the role of system Xc- cystine/glutamate antiporter in the progression of anoxic depolarization (AD) caused by oxygen deprivation during ischemic stroke. Stroke is a leading cause of death and disability worldwide with limited treatment. AD is induced by the loss of membrane potential in neurons due to a cascade of cellular events including an increase of glutamate release, leading to excitotoxicity and neuronal death. System Xc- transporter mediates the exchange of extracellular cystine and intracellular glutamate across the cellular plasma membrane and thereby contributes to most of the tonic glutamate release. With electrophysiological recordings from hippocampal slices in an *in vitro* anoxia model, the authors showed that genetic deletion (with xCT knockout mice) or pharmacological inhibition (with a compound CPG) of system Xc- increased AD latency and altered AD waves. By manipulating the extracellular glutamate concentration or blocking glutamate receptors, they further demonstrated that the protective effect of system Xc- deletion was likely attributed to a reduced level of tonic glutamate that could activate NMDA receptors and accelerate AD. Overall, the experiments were well designed and carefully executed. The findings are novel and have clinical implications. However, the manuscript can be improved if certain results are better interpreted and discussed.

Major comments:

1. In Figure 1 A and 1B, the amplitude of fEPSP at the baseline appeared comparable between the WT and xCT^{-/-} groups. However, the fiber volley preceding the xCT^{-/-} fEPSP was much bigger. Is there a group difference in the amplitude of fiber volley? If yes, this needs to be clarified because the change in the fiber volley is one of the critical signs for AD, as the authors pointed out. If there is no group difference, the authors may choose better representative traces.
2. In Figure 7K, the AD wave duration was not different between the WT and xCT^{-/-} groups under the Sham condition. This is contradictory to the result shown in Figure 1G. Was this because the sample size for Figure 7K was too small? If yes, the authors may consider increasing the sample size.
3. Only male mice were used in this study. Please justify the choice of sexes.
4. A prolonged exposure (6-7 hours) to ACSF might deteriorate the overall health of brain slices. It would be helpful to briefly describe quality control used for recordings in the Materials and Methods section.
5. The system Xc- antagonist (CPG) increased AMPA receptor-mediated synaptic potential in WT neurons, but not in xCT^{-/-} neurons (Figure 6). This difference requires more explanation.

Minor comments:

1. In all relevant figures, please change "Sham" to "Vehicle" to be consistent with the text.
2. Page 4, please define the abbreviation of "mEPSC" and "sEPSC".
3. Page 5, remove "of" from "...Committee at of the University...".
4. Page 6, please define "DC" in "...a large slow DC shift...".
5. Page 7, add "the" in the sentence "...without added drug"?

Dear Dr. Toth,

We are grateful for the careful and insightful examination of our manuscript by the reviewers. We have revised the paper in response to the critiques. We hope that you will agree that the revisions satisfy the concerns of the reviewing editor and considerably improve the paper. The original reviews are included below (in Roman type) along with our responses (Bold type) to each point.

**Warmest Regards,
Bradley Stavros Heit**

Referee #1:

In the paper « Tonic extracellular glutamate and ischemia: glutamate antiporter system xc- regulates anoxic depolarization in hippocampus » the authors investigated the involvement of cystine/glutamate antiporter xc- in anoxic depolarization following an anoxic episode. Focusing on the possible involvement of glutamate present tonically in extracellular space in deleterious postischemic mechanisms is an interesting approach.

To this end they used field recording electrophysiologic technique, mutant mice lacking xc-, its specific antagonist and pharmacologic tools.

The authors show that, when subjected to a transient anoxic period, fEPSP recorded in hippocampal slices from both young and aged xCT^{-/-} mice display a longer AD latency, attenuated AD wave amplitude but increased duration compared to WT. Recording in parallel WT and xCT^{-/-} slices exposed to anoxia within the same chamber is a clever approach.

By pharmacologically blocking xCT, they also reinforce the idea of an implication of this antiport by obtaining results similar to those found with xCT^{-/-} mice.

With the experiments on restoration of ambient glutamate in xCT^{-/-} slices or washout of ambient glutamate in WT slices they demonstrate the involvement of tonic extracellular glutamate in AD latency.

1) In normoxic conditions there is apparently no alteration of the probability of vesicular release at the synapse in xCT^{-/-} mice as demonstrated by PPF. It would have been interesting to perform such a test during and after the anoxic episode to be sure that xCT^{-/-} mutation has also no influence on this short term plasticity mechanism in pathologic conditions.

We thank the referee for this salient observation. We also agree with the referee's comment regarding the potential for short-term plastic alterations in xCT^{-/-} slices after the anoxic insult. Under our experimental conditions, the synaptic response declines rapidly during anoxia and we believe that measures of PPF during rapid response decline are unreliable. Furthermore, the synaptic changes are not synchronous in WT and xCT^{-/-} slices. As an alternative approach to this question, we performed an analysis of PPF magnitude after 45 minutes of recovery for both genotypes. The dataset used for this analysis was from the original Figure 2 (now Figure 3) as these were unpaired experiments, where both WT and mutant slices were left in the depolarized state for exactly one minute prior to reoxygenation. The magnitude of PPF after anoxic insult did not differ between WT and xCT^{-/-} slices ($t_{12} = 0.4222$, $p = 0.6803$). These data have been included in the following sentence in the Results section:

“Lastly, the magnitude of PPF after the recovery phase did not differ between WT and xCT^{-/-} slices ($t_{12} = 0.4222$, $p = 0.6803$; data not shown) suggesting that xCT mutation does not influence alterations in this short-term plasticity mechanism during the acute phase of ischemic recovery.”

2) In the inserts Fig 1 A, B and I where the illustrations are averages of 5 individual responses, what do we see just before the EPSP: the stimulation artifact or the fiber volley? Same remark for illustrations in Fig 2, 4, 5 and 6. It would have been interesting to see illustrations of responses in Fig 3. After 6-7 hours of incubation followed by anoxia and measurement of AD latency, I am curious to know what state the acute hippocampal slices were in. It must have been hard to get valid fEPSPs. Probably not the easiest part of the investigation. It would be interesting to check the evolution of the post-synaptic response to pre-synaptic fibre volley ratio before and after the anoxic episode to see if, in this model, some deleterious effects of the anoxic episode on the integrity of the neuronal network occur within the following post-anoxic hour.

We apologize for lack of clarity in the insets for Figure 1A, B, and thank the referee for pointing this out. The deflections preceding the fEPSPs are in fact stimulus artifacts, not fibre volleys. The fibre volleys overlap in time with the fEPSPs and the two responses cannot always be resolved. In order to more clearly display the inset waveforms, we have broken up the original Figure 1 into two figures (now Figures 1 and 2). The new Figure 1 displays the expanded waveforms for both genotypes to show more clearly the stimulation artifacts, which appear just before the fEPSPs. Further, we have denoted the artifact with an asterisk, and added the appropriate commentary in the Figure 1 legend: “Magenta asterisk denotes the stimulus artifact.”

We agree with reviewer 1 that comparable slice health is crucial for the interpretation of comparisons between animals, especially when slices are maintained *in vitro* for extended periods (experiments in original Figure 3, now Figure 4). We attempted to address the issue for all experiments (including those with extended incubation) by adopting strict criteria for adequate slice health: Hippocampal slices deemed acceptable for study had to display 1) the capacity to generate a fEPSP at least 4 mV in peak amplitude, 2) fEPSPs had to exhibit a half-width less than 7.0 msec during the baseline period, 3) slices had to show PPF values between 135% and 165% during the baseline period, and 4), throughout the baseline period, amplitudes of individual responses could not deviate greater than ± 0.2 mV from the initial response captured at the start of the baseline period Any slice which did not meet these criteria after the extended incubation (as well as in all other experiments) were excluded from testing. We have added this information to the Methods section, which is stated in the following manner:

“Specific parameters were implemented in determining which hippocampal slices would be used for experimentation. Slices deemed acceptable for measurements had to display 1) the capacity to generate an evoked potential of at least 4.0mV in amplitude, 2) fEPSPs which maintained a halfwidth less than or equal to 7.0ms for the duration of the baseline period, and 3) PPF values between 135% and 165% during the duration of the baseline period. Furthermore, throughout the baseline period, amplitudes of individual responses did not deviate greater than ± 0.2 mV from the initial response captured at the start of the baseline period. This was to verify that evoked responses were stable and not “trending” upward or downward prior to anoxic intervention or drug perfusion. Any slices which did not meet these criteria were excluded from testing.”

Incidentally, had the extended duration been deleterious to the health of the slices, AD latencies would be expected to be shorter, not longer (as observed) than after shorter incubations.

The original Figure 3 (now Figure 4) has been expanded in order to display representative experiments for both genotypes with the attendant response traces for the baseline, anoxic, and recovery periods. As can be seen from the figures, waveforms generated in the extended (6-7 hours; new Figure 4) incubation experiments were not qualitatively different when compared those in the shorter (2-3 hours; new Figure 1) incubation experiments.

In order to further address the concerns for the extended incubation period, we have provided additional data analyses. From the 11 paired extended incubation experiments, 5 WT slices and 6 xCT^{-/-} slices displayed the longer AD latency of the given pair, and were therefore left in the depolarized state for exactly one minute before reoxygenation. Since there was no systematic bias in which slice exhibited the longer AD latency (and shorter time in the depolarized state prior to reoxygenation), we could compare post-anoxic recovery between genotypes in this condition. No differential effect in recovery, however, was observed between genotypes ($t_9 = 0.2035$, $p = 0.8433$). These data have been added to the text in the Results section:

“From the 11 paired wash-out experiments, 5 WT slices and 6 xCT^{-/-} slices displayed the longer AD latency out of the pair, and were thus left in the depolarized state for exactly one minute before reoxygenation. This allowed us to empirically measure and compare post-anoxic recovery between genotypes. No differential effect in recovery, however, was observed after extended incubation ($t_9 = 0.2035$, $p = 0.8433$; data not shown).”

3) Effect of CPG administration (Fig 6): if I have correctly understood, the authors performed PPF (75ms IPI) in WT and xCT^{-/-} slices before and during CPG administration, or sham. In Fig 6 A, B, C and D white circles are averages of the first EPSP of the PPF, red circles of the second one (50% greater). Both are normalized to the initial baseline average.

First, the colors of traces pre/post are the same as for the illustration of the traces of the PPF, it's quite confusing, blue and purple would be more coherent with Illustrations C and F. Secondly, if there is an increase of the amplitude of the first EPSP amplitude (by 25% in WT and 13% in xCT^{-/-} mice) but not of the second, as illustrated in Fig6 A, C, D and F, PPF ratio might be modified. In Fig 6 A traces in the inserts don't give such impression, red ones (second EPSP of the PPF, normalized to initial baseline average) are greater after CPG administration. Please give an explanation. A timelapse evolution of PPF ratio before and during CPG perfusion would help to see more clearly its evolution. If there is a modification please give an explanation. Thirdly, the authors use CPG (S-4-carboxphenylglycine) as system xC⁻ inhibitor. However, CPG is known as a competitive group I metabotropic glutamate receptor antagonist. Is there another, more specific, inhibitor of system xC⁻ to perform the same tests? If not additional experiments with another mGluR I antagonist are necessary, AIDA for instance, to verify or clarify the degree of possible involvement of metabotropic receptors.

We agree with the Referee that the color scheme for the original Figure 6 (now Figure 7) needs to be altered. We have changed the colors of the grouped averages of fEPSPs over

the 2-hour pretreatment period to match those in the bar graphs. With respect to PPF, the values plotted in the graphs are the PPF ratios calculated as the amplitude of the second response (of the paired pulses) at each time point divided by the amplitude of the first response at the same time point, expressed as a percentage – that is, the graphs show that the degree of PPF (red) does not change over time even if the amplitude of the first response (purple) does, such as for WT slices during the CPG condition. We have revised the figure caption to reflect this. For clarity, the following statements have been added to the caption of Figure 7 (previously Figure 6):

“PPF values plotted in the graphs are ratios calculated as the amplitude of the second pulse (of the paired pulses) at each timepoint divided by the amplitude of the first response at the same time point, expressed as percentage.”

“Note: the degree of PPF (red) does not change over time even though the amplitude of the first response (purple) does.”

We appreciate the Reviewer’s astute observations regarding the pharmacology used in this study. We did consider use of sulfasalazine (SAS), a system x_c^- inhibitor with no action on mGluRs, however, SAS (Ryu et al., 2003, *J of Pharmacology and Experimental Therapeutics*) and derivatives of SAS (Cho et al., 2010, *Drug News Perspect*; Gwag et al., 2007, *J Cerebral Blood Flow and Metabolism*) have been shown to inhibit NMDARs in models of ischemia. Because NMDARs have explicitly been shown as the primary drivers of AD (Heit et al., 2021; Fusco et al., 2018), we felt CPG was the best choice for this investigation. To the authors’ knowledge, there is currently no evidence suggesting that an mGluR antagonist by itself can alter AD latency. In our experiments, AD latencies in $xCT^{-/-}$ mice were completely unaffected by CPG treatment. This strongly suggests that blockade of type 1 mGluRs is not the mechanism by which CPG affects AD latency.

We have added the following sentences to the Results section regarding this matter:

“Importantly, CPG was chosen over sulfasalazine (SAS) in this regard, because SAS (Ryu et al., 2003) and derivatives of SAS (Cho et al., 2010; Gwag et al., 2007) have been shown to inhibit NMDARs in models of ischemia. NMDARs have been explicitly revealed as the primary drivers of AD (Heit et al., 2021; Fusco et al., 2018), therefore CPG proved to be the best option for this investigation.”

In our study, we also sought to confirm and extend the findings of Williams and Featherstone, who showed that prolonged incubation (glutamate wash-out) can enhance postsynaptic AMPAR abundance and synaptic strength in WT slices; a finding which they replicated in WT slices pretreated with CPG. As such, we tested whether prolonged incubation and CPG treatment would (independently) influence AD latency for WT slices. Indeed, we found it quite striking that despite the likelihood of increased AMPAR abundance elicited by both conditions, WT slices displayed lengthened AD latencies. As mentioned in the text, increased abundance of postsynaptic AMPARs would tend to accelerate AD rather than delay it. We believe these findings further support the excitatory salience of tonic glutamate in the propagation of AD.

4) As clearly illustrated in Fig 1A, B and Fig 2A, B, D, E, there is systematically (?) a lasting depression of fEPSP amplitudes after the anoxic episode in both WT and $xCT^{-/-}$ mice. But this

lasting depression seems to be less important in xCT^{-/-} mice. If so, additional experiments using others technical approaches (WB of glutamate receptors subunits, their phosphorylated / unphosphorylated forms, before and after anoxia for instance) would be necessary to investigate system xC⁻ involvement on such lasting modifications.

The nature of the depression of synaptic transmission after anoxic challenge in slices is incompletely understood, although it has been seen in many studies. The magnitude of the depression is directly related to the time spent in the depolarized state after AD and prior to reoxygenation; it is minimal if the slice is reoxygenated immediately after AD (Heit, et al., 2021). However, whether it is due to neuronal death or reversible changes like presynaptic failure or postsynaptic receptor down-regulation is unclear. The pace of recovery precludes its analysis by electrophysiological methods in acute slices. However, we did address the possibility that system xC⁻ might participate in the depression in the experiments shown in Figure 3 (original Figure 2). When WT and xCT^{-/-} slices spent the same amount of time in the depolarized state after AD and before reoxygenation, we found that both groups showed similar levels of post-anoxic depression (New Fig. 3I). Therefore, regardless of the mechanisms responsible for the post-hypoxic depression, there is no evidence that system xC⁻ is involved.

5) Pages 4 and 27: the authors present the model of hippocampal slices subjected to oxygen deprivation as an extensively used in vitro model of stroke. To support this assertion they cite three references. The Croning and Haddad 1998 reference is just a comparison of brain slice chamber designs (submerged and interface) for investigations of oxygen deprivation in vitro, interestingly glucose deprivation is also used in some of the experiments, pointing the interest about glucose levels. The second citation is the excellent and complete review of P. Lipton on ischemic cell death in neurons, published in 1999. It's interesting to note that P. Lipton has widely used the oxygen/glucose deprivation as an in vitro model of stroke in his papers. The authors also cite Heit et al. 2021 work, which is the only of the three references where anoxia alone is used...

We appreciate and thank the referee for noting this oversight on our part. The total oxygen deprivation model used in our study has indeed been frequently employed as a model for ischemia; however, we failed to list the most notable studies. In the text, we have included the citations listed below. These investigations measured hippocampal neuronal responsiveness during and after anoxia alone as a model for ischemia/stroke.

Wang, T., Raley-Susman, K.M. , Wang, J., Chambers, G., Cottrell, J.E. & Kass, I.S. (1999). Thiopental attenuates hypoxic changes of electrophysiology, biochemistry, and morphology in rat hippocampal slice CA1 pyramidal cells. *Stroke*, 30 (1999) 2400-2407.

Arrigoni, E., Crocker, A.J., Saper, C.B., Greene, R.W., & Scammell, T.E. Deletion of presynaptic adenosine A1 receptors impairs the recovery of synaptic transmission after hypoxia. *Neuroscience*, 132 (2005) 575-580.

- Coelho, J.E., Rebola, N., Fragata, I., Ribeiro, J.A., de Mendonça, A., & Cunha, R.A.**
Hypoxia-induced desensitization and internalization of adenosine A1 receptors in the rat hippocampus. *Neuroscience*, 138 (2006) 1195-1203.
- Pearson, T. & Frenguelli, B.G.** Adrenoceptor subtype-specific acceleration of the hypoxic depression of excitatory synaptic transmission in area CA1 of the rat hippocampus. *Eur. J. Neurosci.*, 20 (2004) 1555-1565.
- Dale, N., Pearson, T., & Frenguelli, B.G.** Direct measurement of adenosine release during hypoxia in the CA1 region of the rat hippocampal slice. *Journal of Physiology*, 526 (2000) 143-155.
- Fischer, M., Reuter, J., Gerich, F.J., Hildebrandt, B., Hägele, S., Katschinski, D., & Müller, M.** Enhanced hypoxia susceptibility in hippocampal slices from a mouse model of Rett syndrome. *J. Neurophysiol.*, 101 (2009) 1016-1032.

Referee #2:

In this study, Heit et al investigated the role of system Xc- cystine/glutamate antiporter in the progression of anoxic depolarization (AD) caused by oxygen deprivation during ischemic stroke. Stroke is a leading cause of death and disability worldwide with limited treatment. AD is induced by the loss of membrane potential in neurons due to a cascade of cellular events including an increase of glutamate release, leading to excitotoxicity and neuronal death. System Xc- transporter mediates the exchange of extracellular cystine and intracellular glutamate across the cellular plasma membrane and thereby contributes to most of the tonic glutamate release. With electrophysiological recordings from hippocampal slices in an in vitro anoxia model, the authors showed that genetic deletion (with xCT knockout mice) or pharmacological inhibition (with a compound CPG) of system Xc- increased AD latency and altered AD waves. By manipulating the extracellular glutamate concentration or blocking glutamate receptors, they further demonstrated that the protective effect of system Xc- deletion was likely attributed to a reduced level of tonic glutamate that could activate NMDA receptors and accelerate AD. Overall, the experiments were well designed and carefully executed. The findings are novel and have clinical implications. However, the manuscript can be improved if certain results are better interpreted and discussed.

Major comments:

1. In Figure 1 A and 1B, the amplitude of fEPSP at the baseline appeared comparable between the WT and xCT^{-/-} groups. However, the fiber volley preceding the xCT^{-/-} fEPSP was much bigger. Is there a group difference in the amplitude of fiber volley? If yes, this needs to be clarified because the change in the fiber volley is one of the critical signs for AD, as the authors pointed out. If there is no group difference, the authors may choose better representative traces.

We apologize for the confusion surrounding the inserts for Figure 1A, B, and thank the Referee for pointing this out. In order to address the questions regarding the individual responses, we have broken up the original Figure 1 into two figures (now Figures 1 and 2). The new Figure 1 displays the expanded waveforms for both genotypes to more clearly show the stimulation artifact, which appears just before the f EPSP. Further, we have denoted the artifact with an asterisk, and added the appropriate commentary in the figure legend. See also referee 1, comment 2.

2. In Figure 7K, the AD wave duration was not different between the WT and xCT^{-/-} groups under the Sham condition. This is contradictory to the result shown in Figure 1G. Was this because the sample size for Figure 7K was too small? If yes, the authors may consider increasing the sample size.

We thank the Referee for the discerning assessment of our data in this regard. Indeed, the lack of a differential effect for AD duration was likely due to the smaller sample size for this set of experiments. That being said, the principal focus of our overall investigation was to examine the AD latency. Although alterations in AD duration aid in the interpretation of our results, we feel that the presented data is sufficient for elucidating our primary aim. Further, in the original Figure 4 (now Figure 5), which indirectly serves as a sham experiment, we do observe a difference in AD duration, which supports the original observation (new Figure 2).

3. Only male mice were used in this study. Please justify the choice of sexes.

We appreciate the referee for this observation, and agree that investigating the potential for differential effects between sexes is an important endeavor. Our prior study, indeed, identified significant differences in AD latency between male and female WT mice (Heit et al., 2021, Figure 10). Therefore, it is not a simple matter of using both males and females and pooling the results, as this would substantially increase the within-group variance in each experiment. Doing the entire study in both males and females separately was cost-prohibited. We chose to study males because almost all of the relevant experimental literature is composed of studies on male animals and we wanted our study to be comparable. These considerations are now included in the Methods section (see below). We hope, in future experiments, to study sex and hormonal interactions with ischemic mechanisms in both WT and mutant mice.

“Prior studies from our group identified significant differences in AD latency between male and female mice (Heit et al., 2021; Figure 10); therefore, in order to decrease within-group variance in each experiment, only male cohorts were used for the current investigation. Furthermore, we wanted our results to be comparable with the existent experimental literature elucidating the AD phenomenon, which almost exclusively involves males.”

Additionally, due to the extensive experimentation planned at the outset this study, we were forced to decide whether to perform seminal experiments on female mice or aged mice. We chose aged mice, because of the provocative observational data, now published (Verbruggen et al., 2022), which came to our attention during the early phases of our

experiments. Verbruggen and colleagues have shown age-dependent alterations in hippocampal function as a consequence of system x_c^- deletion. We chose to align with current research interests, which identify the effects of chronic system x_c^- loss as a subject of great interest.

4. A prolonged exposure (6-7 hours) to ACSF might deteriorate the overall health of brain slices. It would be helpful to briefly describe quality control used for recordings in the Materials and Methods section.

We have added additional material to the Methods section regarding the empirical criteria used for choosing which slices would be tested for experimentation. With respect to concerns regarding the health of slices after prolonged incubation, we have added the waveform traces for both the control and extended incubation periods. As can be seen from the figures, waveforms generated in the extended (6-7 hours) incubation experiments were not qualitatively different when compared those in the shorter (2-3 hours) incubation experiments.

See also referee 1, comment 2.

5. The system X_c^- antagonist (CPG) increased AMPA receptor-mediated synaptic potential in WT neurons, but not in $xCT^-/-$ neurons (Figure 6). This difference requires more explanation.

This is indeed an important observation, which is displayed in Figure 6 (now Figure 7). We did provide an explanation for this phenomenon in the discussion, however, we have extended this explanation and added appropriate citations in the text. The paragraph now reads as follows:

“Importantly, pre-existing lines of evidence suggest tonic glutamate can act in paracrine fashion to suppress glutamatergic synapse strength by triggering removal of postsynaptic glutamate receptors. In our experiments, two hours of system x_c^- inhibition via CPG treatment produced a significant enhancement of fEPSPs in WT, but not $xCT^-/-$ slices. These data support the notion that xCT -mediated alterations of CA1 synapses involve mGluRs coupled to pathways, which modulate postsynaptic AMPAR abundance (Williams and Feathersrone, 2014). Specifically, ambient glutamate can activate extrasynaptic Group 1 mGluRs, which subsequently elicit a cascade of intracellular events resulting in reduced postsynaptic glutamate receptor expression and altered trafficking (Gladding et al., 2009; Oliet et al., 1997; Palmer et al., 1997). In our CPG experiments, ambient glutamate levels were reduced in WT slices as a consequence of system x_c^- antagonism, thereby inhibiting the activation of mGluRs and hence their removal of postsynaptic AMPARs. CPG treatment therefore resulted in increased AMPAR-mediated synaptic transmission, and yet paradoxically increased AD latencies, further highlighting the excitatory salience of tonic glutamate during oxygen-depletion events.”

Minor Comments:

1. In all relevant figures, please change "Sham" to "Vehicle" to be consistent with the text.

We have complied with this request.

2. Page 4, please define the abbreviation of "mEPSC" and "sEPSC".

These acronyms have been properly defined in the text.

3. Page 5, remove "of" from "...Committee at of the University...".

The text has been corrected.

4. Page 6, please define "DC" in "...a large slow DC shift...".

The acronym has been properly defined in the text.

5. Page 7, add "the" in the sentence "...without added drug"?

The text has been corrected.

Dear Dr Larson,

Re: JP-RP-2022-283880X "Tonic extracellular glutamate and ischemia: glutamate antiporter system xc- regulates anoxic depolarization in hippocampus" by John R Larson, Bradley Stavros Heit, Alex Chu, Abhay Sane, David E Featherstone, and Thomas J Park

I am pleased to tell you that your paper has been accepted for publication in The Journal of Physiology.

NEW POLICY: In order to improve the transparency of its peer review process, The Journal of Physiology publishes online as supporting information the peer review history of all articles accepted for publication. Readers will have access to decision letters, including all Editors' comments and referee reports, for each version of the manuscript and any author responses to peer review comments. Referees can decide whether or not they wish to be named on the peer review history document.

The last Word version of the paper submitted will be used by the Production Editors to prepare your proof. When this is ready you will receive an email containing a link to Wiley's Online Proofing System. The proof should be checked and corrected as quickly as possible.

Authors should note that it is too late at this point to offer corrections prior to proofing. The accepted version will be published online, ahead of the copy edited and typeset version being made available. Major corrections at proof stage, such as changes to figures, will be referred to the Reviewing Editor for approval before they can be incorporated. Only minor changes, such as to style and consistency, should be made a proof stage. Changes that need to be made after proof stage will usually require a formal correction notice.

All queries at proof stage should be sent to TJP@wiley.com.

Are you on Twitter? Once your paper is online, why not share your achievement with your followers. Please tag The Journal (@jphysiol) in any tweets and we will share your accepted paper with our 23,000+ followers!

Yours sincerely,

Katalin Toth
Senior Editor
The Journal of Physiology

P.S. - You can help your research get the attention it deserves! Check out Wiley's free Promotion Guide for best-practice recommendations for promoting your work at www.wileyauthors.com/eeo/guide. And learn more about Wiley Editing Services which offers professional video, design, and writing services to create shareable video abstracts, infographics, conference posters, lay summaries, and research news stories for your research at www.wileyauthors.com/eeo/promotion.

*** IMPORTANT NOTICE ABOUT OPEN ACCESS ***

To assist authors whose funding agencies mandate public access to published research findings sooner than 12 months after publication The Journal of Physiology allows authors to pay an open access (OA) fee to have their papers made freely available immediately on publication.

You will receive an email from Wiley with details on how to register or log-in to Wiley Authors Services where you will be able to place an OnlineOpen order.

You can check if you funder or institution has a Wiley Open Access Account here: <https://authorservices.wiley.com/author-resources/Journal-Authors/licensing-and-open-access/open-access/author-compliance-tool.html>.

Your article will be made Open Access upon publication, or as soon as payment is received.

If you wish to put your paper on an OA website such as PMC or UKPMC or your institutional repository within 12 months of publication you must pay the open access fee, which covers the cost of publication.

OnlineOpen articles are deposited in PubMed Central (PMC) and PMC mirror sites. Authors of OnlineOpen articles are permitted to post the final, published PDF of their article on a website, institutional repository, or other free public server, immediately on publication.

Note to NIH-funded authors: The Journal of Physiology is published on PMC 12 months after publication, NIH-funded authors DO NOT NEED to pay to publish and DO NOT NEED to post their accepted papers on PMC.

EDITOR COMMENTS

Reviewing Editor:

The authors have done an excellent job of responding to the reviewers comments with no further concerns.

REFEREE COMMENTS

Referee #1:

The responses and explanations provided by the authors to the comments made to them are satisfactory and correctly documented when necessary.

The complementary experiments requested have been carried out seriously and provide additional valuable information. The whole brings more clarity and solidity to this work.

Referee #2:

With meticulous experiments, the authors have revealed a new role of system Xc- cystine/glutamate antiporter in the progression of ischemic stroke. They have addressed all my previous concerns. The revised manuscript is improved significantly.

1st Confidential Review

27-Sep-2022